# INDUCTION: Finite-Structure Concept Synthesis in First-Order Logic

**Serafim Batzoglou** [1]

## Abstract

Induction is the search for a general rule that explains observations. We study logical induction in finite relational worlds: each problem gives small structures over a fixed vocabulary, labels objects belonging to an unknown unary concept, and asks for one first-order formula $\varphi(x)$ that accounts for those labels across worlds. Finite domains make formulas mechanically checkable by exact evaluation and SMT. We introduce INDUCTION, a benchmark for finite-structure concept synthesis with three regimes: FULLOBS (full observation), where all facts are observed; CI (contrastive induction), where YES/NO worlds require discriminative hypotheses; and EC (existential completion), where validity is defined by world-local completion of unknown facts. We evaluate frontier language models, include symbolic synthesis baselines, and score both validity and formula size. Prompted models show real but incomplete capability, with sharp difficulty gradients and hard structural families. Held-out evaluation shows that compact formulas generalize far better than bloated ones; parsimony separates concept recovery from finite-world fit.

## 1. Introduction

Much of logic evaluation emphasizes deduction from a given specification. A complementary challenge is induction: can a system discover a compact rule from observations? This ability is central to scientific and mathematical reasoning, where success is not merely fitting examples, but finding simple hypotheses that remain stable under new evidence. Recent language and reasoning models can emit well-formed logical expressions, including first-order formulas (Polu & Sutskever, 2020; Tafjord et al., 2021; Han et al., 2024), but it remains unclear whether they can reliably synthesize compact first-order definitions from relational

evidence rather than overfit the observed cases.

We study this question through *finite-structure concept synthesis*. Each problem contains several finite relational worlds over a fixed signature, together with object-level labels indicating which elements belong to an unknown unary concept. The solver sees these labels extensionally and must output one first-order formula $\varphi(x)$ whose extension matches the labeled concept across all worlds. Thus the task asks for an intensional explanation of extensional data: a single symbolic hypothesis that works uniformly across multiple structures. Because the domains are finite, every candidate formula can be evaluated exactly by finite-model checking and SMT, making correctness independent of natural-language interpretation.

This setting connects to inductive logic programming, relational concept learning, and constraint-based synthesis (Muggleton, 1991; Quinlan, 1990; Alur et al., 2013), but the benchmark interface is deliberately simple and fully extensional: the input is a collection of finite worlds and target labels, and the output is a formula in a fixed first-order language. This puts prompted language/reasoning models and symbolic baselines under the same mechanically checkable semantics.

We introduce INDUCTION, a benchmark suite with three complementary regimes. In FULLOBS (full observation), all predicate facts are observed and the formula must match the target extension in every input world. In CI (contrastive induction), YES worlds must be matched exactly, while NO worlds must not be matched exactly. In EC (existential completion), some non-target facts are unknown, and validity is defined by world-local existential completion. These regimes probe multi-world induction, use of negative evidence, and reasoning under missing information.

Since finite evidence can admit many consistent formulas, validity alone is not enough for evaluation. A model can sometimes satisfy the input constraints with a large case-splitting formula that exploits accidental regularities of the finite worlds. We therefore report both unbounded success and budgeted metrics based on formula size. Held-out evaluation shows why this matters: among instance-correct solutions, compact near-gold formulas generalize far better than bloated ones, so parsimony is tied to whether the returned formula captures the intended concept.

We emphasize controlled difficulty. Gold concepts are

[1]Independent Researcher. Correspondence to: Serafim Batzoglou <serafim.batzoglou@gmail.com>.

*Proceedings of the 43rd International Conference on Machine Learning*, Seoul, South Korea. PMLR 306, 2026. Copyright 2026 by the author(s).

drawn from a bounded structural template library, worlds are constructed to eliminate plausible shortcut hypotheses, and rejection filters remove instances solved by trivial atomic, subformula, or quantifier-free alternatives. For CI, targeted trap construction makes NO worlds informative by killing shortcuts that survive the YES worlds. These mechanisms make the benchmark diagnostic: failures can be traced to structural properties such as universal relational generalization, lifted $x$-relations, or case-splitting bloat.

Our contributions are:

- We formalize finite-structure concept synthesis as a solver-verifiable first-order induction task and instantiate it in three regimes: FULLOBS, CI, and EC.
- We construct a controlled v1 benchmark using bounded target libraries, distractor-guided world generation, contrastive traps, and shortcut-rejection filters.
- We evaluate frontier language/reasoning models together with symbolic synthesis baselines, using exact semantics and formula-size-aware metrics.
- We show that compactness predicts generalization: among correct formulas, near-gold solutions generalize much better to held-out worlds than bloated case-splitting solutions.
- We release the datasets and generation/evaluation code to support reproducible study of symbolic induction.

## 2. Problem Setup

**Finite relational worlds.** All tasks take place in finite first-order structures. We fix a small relational signature $\Sigma = \{P, Q, R, S\}$, with unary predicates $P(\cdot), Q(\cdot)$ and binary predicates $R(\cdot, \cdot), S(\cdot, \cdot)$. A world $W$ is a finite $\Sigma$-structure with domain $D_W = \{a_1, \ldots, a_n\}$ and interpretations $P^W, Q^W \subseteq D_W$ and $R^W, S^W \subseteq D_W \times D_W$. Each world also has a unary target extension $T_W \subseteq D_W$, given extensionally as the set of objects labeled positive; objects outside $T_W$ are labeled negative. The goal is to recover an intensional definition of this target: a single first-order formula $\varphi(x)$ that explains the labels across multiple worlds. The same formula is evaluated in every world, rather than fitting a different formula per world.

**Instances.** A benchmark instance $I$ consists of a finite set of worlds $\{W_1, \ldots, W_k\}$ sharing the signature $\Sigma$, but possibly with different domains and predicate interpretations. A system must output one first-order formula $\varphi(x)$ with exactly one free variable $x$. Given a world $W$, the formula induces the predicted extension

$$\widehat{T}_W(\varphi) = \{a \in D_W \mid W \models \varphi[a]\}. \quad (1)$$

The task variants in Section 3 specify how these predicted extensions are compared with the target extensions $T_W$: exact matching in FullObs, contrastive YES/NO matching in CI, and matching under existential completion in EC.

**Symbolic output language.** Systems output $\varphi(x)$ in a Lisp-style S-expression syntax over $P, Q, R, S$, equality, Boolean connectives, and first-order quantifiers. The output must have exactly one free variable, $x$; all other variables must be bound. Equality atoms such as $(= x \, y)$ are parsed and evaluated under standard semantics. Appendices C–E provide the output grammar, qualitative examples, and task prompts. Outputs are parsed into an AST and checked mechanically.

**Model checking and solver-verifiable semantics.** Because all domains are finite, evaluation reduces to finite model checking (Torlak & Bodik, 2014; Alur et al., 2013). Our implementation evaluates a candidate formula by grounding quantifiers over the finite domain and translating the result to a Boolean circuit, with Z3 used for consistency checks. Thus correctness is mechanically checkable and independent of any natural-language interpretation.

**Complexity measures and budgeted scoring.** A formula may satisfy the input constraints while being much larger than the intended concept, for example by case-splitting over accidental regularities of the finite worlds. To distinguish compact concept recovery from such bloat, we track two syntactic measures: AST size, the number of nodes in the parsed formula tree, and quantifier depth (QD), the maximum nesting depth of quantifiers. Quantifier depth is quantifier rank, not the number of $\forall/\exists$ alternations.

Many instances are generated from a planted gold formula $\varphi^\star$. We therefore report gold-relative success rates such as Acc@gold+$\Delta$: the fraction of instances solved with $\mathrm{AST}(\widehat{\varphi}) \leq \mathrm{AST}(\varphi^\star) + \Delta$. The planted formula is a reference complexity anchor, not a claim of global minimality in the full output language. We also report bloat rate, e.g., $\mathrm{AST}(\widehat{\varphi}) > \mathrm{AST}(\varphi^\star) + 25$, to quantify how often correctness relies on highly non-canonical solutions.

## 3. Induction Tasks

All three tasks use the same objects: input worlds, target extensions, and a single output formula $\varphi(x)$. We refer to the tasks as FULLOBS (full observation), CI (contrastive induction), and EC (existential completion). They differ in which facts are shown to the system and in the semantic condition for validity of $\varphi(x)$. We write $I = \{W_1, \ldots, W_k\}$ for the worlds in an instance.

### 3.1. Full Observation across Multiple Worlds

In FullObs, each input world provides a complete interpretation of the observed predicates $P, Q, R, S$ over its finite domain. A candidate $\varphi(x)$ solves an instance iff its predicted extension matches the target extension in every world:

$$\forall W \in I : \widehat{T}_W(\varphi) = T_W. \quad (2)$$

This is the direct multi-world concept-synthesis setting: the challenge is to find one relational definition that explains the labeled objects across several finite structures.

## 3.2. Contrastive Induction

In CI, the worlds are partitioned into YES worlds $I_{\text{YES}}$ and NO worlds $I_{\text{NO}}$. YES worlds impose the same exact-match constraint as FullObs. NO worlds provide world-level contrastive evidence: a valid formula must *not* exactly match their target extensions. Formally, define

$$\text{MATCH}(W, \varphi) \Longleftrightarrow \widehat{T}_W(\varphi) = T_W. \tag{3}$$

A candidate $\varphi(x)$ solves a CI instance iff it matches every YES world and does not exactly match any NO world:

$$\begin{aligned}(\forall W \in I_{\text{YES}} : \text{MATCH}(W, \varphi)) \wedge \\ (\forall W \in I_{\text{NO}} : \neg\text{MATCH}(W, \varphi)).\end{aligned} \tag{4}$$

Thus CI does not ask $\varphi$ to invert the labels on NO worlds; it only asks that $\varphi$ fail to be an exact explanation of each NO target extension. During generation, NO worlds are constructed to make this contrastive condition informative by targeting shortcut formulas that survive the YES worlds.

## 3.3. Partial Observation and Existential Completion

The EC variant introduces missing information in the non-target predicates. For each world $W$, ground atoms of $P, Q, R, S$ are partitioned into known true, known false, and unknown. Let $\Omega_W$ denote the unknown ground atoms in $W$. A completion $C$ assigns a truth value to every atom in $\Omega_W$, yielding a fully specified world $W^C$. The target extension $T_W$ remains observed.

Under existential-completion semantics, a formula is valid if, for each world independently, there exists some completion under which it matches the target:

$$\forall W \in I, \ \exists C_W \in \text{Comp}(W) : \ \widehat{T}_{W^{C_W}}(\varphi) = T_W. \tag{5}$$

Completions are world-local: each world may use a different assignment to its unknown atoms. Equivalently, $\varphi$ must be compatible with at least one completion of the observed facts and labels in each world. EC is therefore existential, not universal, over completions.

EC evaluation introduces Boolean variables for unknown atoms, grounds quantifiers over the finite domain, and asks a solver whether the resulting constraints are satisfiable.

# 4. Dataset Generation

A central goal of INDUCTION is diagnostic, controllable difficulty. Instances should not be trivial, but they should also not be opaque hard cases. Our generator therefore separates two choices: first, it defines a bounded pool of target formulas with controlled structural coverage; second, it constructs finite worlds that make the chosen target informative by eliminating plausible shortcut hypotheses. Rejection filters remove degenerate instances solved by simple atomic, subformula, or quantifier-free alternatives. We describe the target pool and task-specific world generation below; Appendices B.12 and C.1 give generator details and the target-pool census.

## 4.1. Gold Formula Pool and Structural Tagging

Gold target concepts $\varphi^\star(x)$ are drawn from a curated template pool comprising approximately 200 structurally distinct formulas. Each formula is tagged by family, based on quantifier structure such as single-$\exists$, single-$\forall$, nested $\exists\forall$, and nested $\forall\exists$, and by subfamily, a finer signature capturing guard literals, relation usage, and nesting patterns. By a guard we mean a literal that restricts a quantified witness, e.g., $P(y)$ in $\exists y\,(P(y) \wedge \cdots)$.

**Target-pool construction and exhaustiveness.** The target language is bounded by this structural template library rather than by exhaustive syntactic enumeration of all formulas under a size/depth bound. Syntactic enumeration would overrepresent surface variants such as renamings, Boolean rewrites, argument-order variants, and mechanically generated predicate variants, rather than giving balanced diagnostic coverage. Within the bounded target language, however, candidate-target enumeration is exhaustive at the eligible-pool level: we instantiate templates with predicate and argument-order variants, deduplicate the resulting formulas, apply task- and band-specific filters, and enumerate the eligible pool before selecting planted gold formulas. Thus the benchmark is not exhaustive over bounded FO syntax; rather, each task/band enumerates the eligible target pool induced by the bounded target language before balanced gold selection and world construction. Appendix C.1 gives the algorithm and pool-size census.

Gold formulas are then selected per band with constraints on quantifier depth, AST size, family, and subfamily distribution. To ensure variety, we use balancing caps and quotas so that no single formula, family, or subfamily dominates; in some bands the same formula seeds multiple instances with different generated worlds. Gold formulas are controlled reference concepts from this bounded target language, not canonical representatives of equivalence classes.

Although equality is allowed in the output language, our v1 planted gold templates do not use it. This design keeps the target distribution focused on relational induction—predicates and quantifiers—rather than identity-based constraints. Systems are free to use equality in predictions, and such formulas are evaluated normally.

A particularly important subfamily involves *lift-hard* patterns: formulas where a relation involving the free variable $x$ appears inside a universally quantified scope. For example,

$$\forall y\,(R(x, y) \to \exists z\, S(y, z))$$

requires checking a property of all $R$-successors of $x$. These patterns frequently expose failures of universal relational generalization, so we use them to create harder diagnostic slices in FullObs and CI.

## 4.2. FullObs Generation

FullObs instances require a candidate formula to match $T$ across $k$ fully observed input worlds. The challenge is constructing worlds that make the gold concept *highly constraining*—i.e., that eliminate a large fraction of plausible shortcut or structurally different hypotheses, even though the gold formula is not guaranteed to be uniquely identified by the finite worlds.

**Hypothesis pool.** We maintain a frozen pool of $\approx 1{,}500$ candidate hypotheses organized into three tiers: (i) simple shortcuts (atomic predicates, QD = 1 formulas with AST $\leq 10$); (ii) near-miss mutants of gold formulas (predicate swaps, guard drops, argument permutations); (iii) structurally complex distractors (QD = 2, larger AST).

**World generation with kill tracking.** For each gold formula $\varphi^\star$, each candidate world is a random finite model with a target extension induced by $\varphi^\star$. Unary predicates $P, Q$ are sampled via Bernoulli trials with $p_{\text{unary}} = 0.4$ (constrained to 15–85% true per predicate). Binary predicates $R, S$ use regular out-degree sampling: each element has exactly 2 outgoing edges for each relation to keep relational density stable across domain sizes and avoid degenerate sparse/dense regimes for quantifiers. Domain sizes vary by band (5–7 for easy, 7–10 for medium, 8–12 for hard). Before accepting a world, we require that it kills at least one surviving hypothesis from the pool, i.e., a hypothesis that matched all previous worlds but fails on this one.

**Filters.** After generating $k$ worlds, we apply rejection filters (Appendix Table 28): (i) **atomic**: reject if any atomic formula matches $T$ in all worlds; (ii) **subformula**: reject if any proper subformula of $\varphi^\star$ matches all worlds; (iii) **quantifier-free**: reject if any quantifier-free combination matches all worlds. These filters remove the most obvious non-quantified shortcuts and make accepted instances more likely to require genuine quantifier reasoning.

**Band configuration.** FullObs v1 uses six bands with increasing difficulty: **simple** ($k = 4$, QD = 1), **easy** ($k = 6$, QD $\geq 2$), **medium** ($k = 8$), **hard** ($k = 10$, tighter filters), and two extreme slices, extreme_logic and extreme_context, with $k = 10$ and lift-hard formulas.

## 4.3. CI Generation: Tiered Trap Pools

CI instances require a candidate formula to match $T$ on all YES worlds while not exactly matching any NO world. The key insight is that NO worlds should be *informative*—they should rule out plausible shortcut hypotheses that a system might choose after seeing only the YES worlds.

**Trap-based contrastive construction.** For each gold formula $\varphi^\star$, we construct a per-problem *trap pool* of plausible alternatives: simple shortcuts (AST $\leq 10$, QD $\leq 1$) and near-miss mutants of $\varphi^\star$ (guard drops, predicate swaps, argument permutations). YES worlds are generated incrementally, tracking which traps *survive*—i.e., match $T$ on all

YES worlds so far. We enforce a *tight survivor band*: after all YES worlds, the number of surviving near-miss traps should be in $[1, 2]$ and total survivors in $[2, 4]$. This band was calibrated empirically: too many survivors make matching YES worlds with shortcuts too easy; too few make NO worlds uninformative. Each NO world is then constructed so that at least one surviving trap exactly matches the NO target while the planted gold does not.

**Band configuration.** CI v1 uses two bands: **core** (no lift-hard, 7–8 YES worlds, 2–3 NO worlds) and lift_mix (35% lift-hard formulas, same world counts). The lift_mix band remains less saturated for systems that master the core band.

## 4.4. EC Generation: Partial Observation

EC instances present worlds where some non-target ground atoms are *unknown*: their truth values are hidden from the system. A candidate formula $\varphi$ is valid if, for each world, there exists a *completion* of the unknown atoms under which $\varphi$ matches the observed target labels exactly. This existential-completion semantics is checked via Z3: we ground quantifiers over finite domains and search for a satisfying assignment to the unknowns.

**Unknown atom selection.** For each world, we designate a subset of ground atoms as unknown. In v1, 20% of binary atoms are masked as unknown: the core band masks atoms from $R$ and $S$, while the hard band masks only $R$ atoms, leaving $P, Q, S$ fully observed. The system observes only the known atoms; unknown atoms are explicitly marked as unknown in prompted evaluations. We also apply relevance filters so that masked atoms are not merely decorative; Appendix B.13 gives the exact filters.

**Band configuration.** EC v1 uses two bands: **core** (120 problems): $k = 3$ worlds, domain sizes 6–8, QD = 1 gold formulas, 20% unknown rate on $R, S$. **hard** (80 problems): $k = 3$ worlds, domain sizes 7–9, QD = 2 gold formulas, 20% unknown rate on $R$ only. The core band is designed to be solvable by multiple systems, while the hard band remains difficult for many current systems.

# 5. Evaluation and Results

**Metrics and reporting protocol.** We report all headline metrics with denominator equal to all instances. For prompted models, missing or unparsable outputs count as incorrect; coverage is the fraction of instances for which a parseable formula is returned. FullObs and CI report task success under their exact-match or contrastive semantics, while EC reports existential-completion validity. We also report budgeted success, Acc@gold+$\Delta$, which requires both task success and $\text{AST}(\hat{\varphi}) \leq \text{AST}(\varphi^\star) + \Delta$, and bloat rate, the fraction of successful formulas with $\text{AST}(\hat{\varphi}) > \text{AST}(\varphi^\star) + 25$. Prompted models in Table 1 use the same prompt template and standardized decoding limits; model identifiers and decoding settings ap-

pear in Appendix B.14. Symbolic baselines in Table 2 are bounded searches under the same exact task semantics, with timeout/no-solution counted as incorrect.

**Across-task summary.** Table 1 summarizes prompted-model performance across FullObs, CI, and EC. Across tasks, unbounded and budgeted success can diverge substantially, especially in FullObs and EC. To test whether formula bloat reflects harmless syntactic redundancy or overfitting, we evaluate held-out worlds labeled by the planted gold concept. The resulting generalization curves (Tables 3 and 6; Figure 2) show that compact, low-bloat formulas generalize much better than high-bloat formulas, supporting gold-relative budgets as a proxy for conceptual abstraction rather than case-by-case patching.

**Symbolic calibration.** Table 2 calibrates INDUCTION beyond prompted language models. The unified SMT baseline is strong on CI and EC, but substantially weaker on FullObs; a more engineered FullObs-specific Z3 portfolio closes much of that gap. Thus the benchmark is not solved by a single generic symbolic baseline. These rows are bounded searches under a fixed compute budget, not upper bounds on symbolic performance.

No single prompted model dominates all three induction tasks. Grok4 is strongest on FullObs when scored over all problems (50.7% Acc all, 46.7% Acc@gold+25) with 89.3% coverage, Gemini 3.5 Flash has the best budgeted CI performance (78.0% Acc@gold+25), GPT-5.4 clearly leads EC (93.5% validity, 64.0% Acc@gold+25), and GPT-5.2 retains the best raw CI accuracy (82.5%). Relative to GPT-5.2, GPT-5.4 improves budgeted performance while reducing bloat in both FullObs and CI, making the version-to-version gain one of better parsimony rather than higher unbounded accuracy alone.

### 5.1. FullObs: difficulty, bloat, and generalization

FullObs v1 contains 375 problems across six bands: simple (QD=1), easy, medium, hard, and two extreme slices, extreme_logic and extreme_context (Appendix C.1). Figure 1 shows budgeted accuracy curves, and Appendix B gives additional diagnostic breakdowns.

FullObs exposes the core multi-world induction challenge.

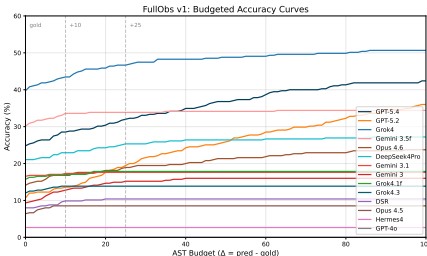

*Figure 1.* FullObs v1 budgeted accuracy curves: Acc@(+Δ) for Δ ∈ [0, 100].

The simple QD=1 slice is near-saturated for several frontier models, but performance drops sharply once formulas require QD=2 relational structure and more worlds. Grok4 remains strongest when scored over all problems, reaching 50.7% Acc all and 46.7% Acc@gold+25, but its 89.3% coverage still limits reliability relative to full-coverage systems. Among near-full-coverage systems, GPT-5.4 and GPT-5.2 have the highest unbounded accuracy (43.5% and 43.7%), while Gemini 3.5 Flash has the strongest budgeted score (33.9% Acc@gold+25) and very low bloat (0.5%). GPT-5.4 remains much stronger than GPT-5.2 under budgeted scoring (32.0% vs. 19.5% Acc@gold+25) and cuts bloat from 24.3% to 11.5%. The band breakdown sharpens this contrast: GPT-5.2 is best on the simplest slices (100.0% simple, 89.0% easy), whereas GPT-5.4 is stronger on medium, hard, and extreme problems (39.0/25.0/8.0 vs. 29.0/21.0/0.0), suggesting better robustness once the required structure becomes less local. Some systems also use equality, although the planted gold templates do not; Appendix B.15 summarizes equality usage and success rates.

**Bloat hurts generalization.** To test whether bloated formulas capture the intended concept or merely fit the input worlds, we generated $H = 5$ held-out worlds per FullObs instance, drawn IID from the same generator distribution and labeled by the planted gold formula $\varphi^\star$. We report held-out exact match: the fraction of held-out worlds where the predicted formula matches $\varphi^\star$'s extension on all domain elements. Table 3 shows that, among instance-correct formulas, near-gold solutions (AST $\leq$ gold+1) generalize at 77.3–97.6%, whereas above-gold solutions drop to 6.7–73.3%. The effect is especially pronounced for strong models: GPT-5.4 falls from 92.6% to 24.8%, GPT-5.2 from 88.7% to 15.9%, and DeepSeek4Pro from 96.6% to 30.4%, when moving from near-gold to above-gold formulas. Figure 2 shows the same monotone trend by AST-delta bin. Thus bloat is not merely cosmetic: it is strongly associated with overfitting to the finite input instance.

**Controlling for instance difficulty.** One possible confound is that bloated formulas might occur on intrinsically harder problems. We control for this by comparing multiple instance-correct predictions for the same problem across systems. For each problem with at least two correct predictions, we compare held-out generalization of the shortest and longest formulas, excluding exact gold matches, which have perfect held-out generalization by construction. Table 4 shows that shorter formulas generalize better within the same problem: they achieve 86.9% held-out exact match versus 12.5% for bloated formulas, with positive gaps on 92% of problems and negative gaps on only 3%. This within-problem control supports bloat-aware scoring: compactness is a reliable signal of conceptual generalization.

*Table 1.* **Across-task v1 summary (snapshot).** FULLOBS (full observation), CI (contrastive induction), and EC (existential completion). *Acc_all* (exact-match accuracy) with denominator=*all* instances (missing or unparsable outputs count as incorrect). Acc@gold+25 = budgeted accuracy with $\mathrm{AST}(\hat{\varphi}) \leq \mathrm{AST}(\varphi^{\star}) + 25$. Cov = coverage (fraction with parseable formula); Parse = parse error rate. Gemini 3.5f denotes Gemini 3.5 Flash.

| Model | FullObs (375) | | | | CI (200) | | | | EC (200) | | | |
|---|---|---|---|---|---|---|---|---|---|---|---|---|
| | Acc | @+25 | Cov | Parse | Acc | @+25 | Cov | Parse | Acc | @+25 | Cov | Parse |
| GPT-5.4 | 43.5% | 32.0% | 100.0% | 0.0% | 79.0% | 76.0% | 100.0% | 0.0% | **93.5%** | **64.0%** | 99.5% | 0.0% |
| GPT-5.2 | 43.7% | 19.5% | 100.0% | 0.0% | **82.5%** | 73.0% | 100.0% | 0.0% | 78.0% | 59.5% | 100.0% | 0.0% |
| Grok4 | **50.7%** | **46.7%** | 89.3% | 0.0% | 78.0% | 75.5% | 100.0% | 0.0% | 53.0% | 53.0% | 99.5% | 0.0% |
| Gemini 3.5f | 34.4% | 33.9% | 99.7% | 0.3% | 78.5% | **78.0%** | 100.0% | 0.0% | 51.5% | 51.0% | 100.0% | 0.0% |
| Opus 4.6 | 23.7% | 18.9% | 100.0% | 0.0% | 79.5% | 75.5% | 100.0% | 0.0% | 58.0% | 57.0% | 99.5% | 0.0% |
| DeepSeek4Pro | 28.0% | 25.3% | 99.2% | 0.0% | 66.5% | 63.5% | 98.5% | 0.0% | 53.5% | 50.5% | 99.5% | 0.0% |
| Gemini 3.1 | 17.6% | 17.6% | 100.0% | 0.0% | 70.0% | 70.0% | 100.0% | 0.0% | 53.0% | 53.0% | 100.0% | 0.0% |
| Gemini 3 | 16.0% | 15.2% | 100.0% | 0.0% | 55.0% | 55.0% | 100.0% | 0.0% | 53.5% | 52.0% | 100.0% | 0.0% |
| Grok4.1f | 17.9% | 17.9% | 98.4% | 1.6% | 60.5% | 60.5% | 98.0% | 2.0% | 41.0% | 41.0% | 98.5% | 1.5% |
| Grok4.3 | 13.9% | 13.9% | 91.2% | 0.0% | 46.0% | 45.0% | 100.0% | 0.0% | 38.5% | 38.5% | 100.0% | 0.0% |
| DSR | 10.4% | 10.4% | 99.7% | 0.3% | 41.5% | 41.5% | 99.0% | 0.5% | 33.0% | 33.0% | 100.0% | 0.0% |
| Opus 4.5 | 8.5% | 8.5% | 98.7% | 1.3% | 34.0% | 34.0% | 100.0% | 0.0% | 30.0% | 30.0% | 99.0% | 1.0% |
| Hermes4 | 2.7% | 2.7% | 99.5% | 0.5% | 2.5% | 2.5% | 99.5% | 0.5% | 15.5% | 15.5% | 100.0% | 0.0% |
| GPT-4o | 0.0% | 0.0% | 100.0% | 0.0% | 0.5% | 0.5% | 99.0% | 1.0% | 2.0% | 2.0% | 100.0% | 0.0% |

*Table 2.* Symbolic baselines. All outputs are rechecked by the benchmark evaluator under the exact task semantics. Timeout/no-solution counts as incorrect; timeout is 30 minutes per instance.

| Method | FullObs | CI | EC |
|---|---|---|---|
| z3-prenex + rescue | 48.8% | 82.0% | 100.0% |
| z3-ad-mix | 65.9% | -- | -- |
| ilasp-frag | 15.7% | 45.5% | -- |

*Table 3.* **Held-out generalization by formula complexity (FullObs).** Near-gold: AST ≤ gold+1; above-gold: AST > gold+1. $\Delta$ = near-gold − above-gold.

| Model | #Valid | Near-Gold Gen% | Above-Gold Gen% | $\Delta$ |
|---|---|---|---|---|
| Grok4 | **181** | **97.6%** | 41.1% | +56.6% |
| GPT-5.4 | 156 | 92.6% | 24.8% | +67.8% |
| GPT-5.2 | 156 | 88.7% | 15.9% | +72.8% |
| Gemini 3.5f | 123 | 93.6% | 56.9% | +36.7% |
| DeepSeek4Pro | 101 | 96.6% | 30.4% | +66.2% |
| Opus 4.6 | 84 | 86.8% | 20.6% | +66.2% |
| Grok4.1f | 60 | 93.7% | 6.7% | +87.0% |
| Gemini 3.1 | 59 | 84.6% | **73.3%** | +11.3% |
| Gemini 3 | 53 | 77.3% | 35.7% | +41.7% |
| Grok4.3 | 46 | 91.7% | 52.0% | +39.7% |

## 5.2. CI: contrastive success and shortcut failures

CI v1 contains 200 problems with two bands: core (120) and lift_mix (80). Table 5 decomposes outcomes into correct solutions, YES-failures, NO-failures, parse errors, and missing responses. Here a YES-failure means the formula does not match a YES world, while a NO-failure means it exactly matches a contrastive NO target and therefore violates the CI non-match condition. Thus NO-failure is not label inversion; it is failure to avoid a shortcut target.

Most CI failures occur on YES worlds, but CI instances contain many more YES worlds than NO worlds (7.9 vs. 2.0), so this does not by itself indicate a special weakness on positive evidence. Rather, the decomposition shows how models fail to use the contrastive information. The strongest systems make very few NO-failures: GPT-5.2 has

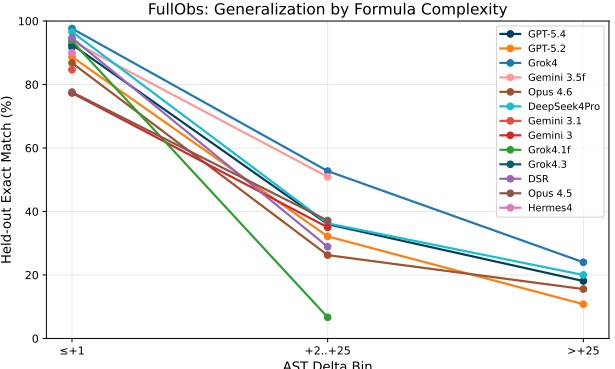

*Figure 2.* FullObs held-out generalization by AST delta bin. For instance-correct predictions, we measure exact-match rate on held-out worlds. Bins with at most five predictions are omitted. Generalization degrades monotonically with formula complexity across the retained points. Curves show selected prompted models; tables give full results.

only 0.5%, GPT-5.4 and Opus 4.6 1.0%, Gemini 3.5 Flash 1.5%, and Grok4 2.5%; weaker systems also miss YES worlds or match NO shortcut targets.

CI is also sensitive to generation design. The trap generator deliberately preserves plausible shortcuts through the YES worlds and then constructs NO worlds that kill those shortcuts. This makes NO worlds informative, but it also means the accepted YES worlds can be easier to satisfy with simple formulas. Accordingly, CI raw accuracy is often higher than FullObs accuracy, and should be interpreted together with the failure decomposition and held-out analysis rather than as a direct measure of concept recovery.

CI reveals a different tradeoff from FullObs. GPT-5.2 retains the best raw CI accuracy (82.5%), while Gemini 3.5 Flash attains the best budgeted CI score (78.0% Acc@gold+25). GPT-5.4 remains a useful version-to-version comparison: relative to GPT-5.2, it trades a small drop in raw CI exact-

match for more compact solutions, reducing bloat from 9.5% to 3.0% while improving Acc@gold+25 from 73.0% to 76.0%. Grok4 is also instructive: it has competitive raw and budgeted accuracy with no parse or missing failures, but makes slightly more NO-failures than the top GPT/Opus/Gemini group.

Table 6 reports held-out generalization for CI, using 3 YES and 2 NO held-out worlds per instance. A key subtlety is that CI correctness does not uniquely identify the planted gold concept. Many formulas can match the finite YES worlds while avoiding exact matches on the input NO worlds without coinciding with $\varphi^\star$. Consequently, absolute held-out YES exact-match rates are low even among instance-correct CI solutions. Nevertheless, the same qualitative pattern persists as in FullObs: near-gold CI solutions generalize better than above-gold ones, and the generalization rate decays monotonically with AST delta (Appendix B.9). DeepSeek4Pro shows the largest near-gold advantage in Table 6 (+14.6 points), followed by Gemini 3.5 Flash (+13.4), GPT-5.4 (+10.2), and Grok4 (+8.8). Thus the bloat effect is weaker in CI than in FullObs, but still visible.

### 5.3. EC: completion validity and the parsimony gap

EC v1 contains 200 problems with two bands: core (120; QD=1) and hard (80; QD=2). Table 7 shows the core/hard split, and Figure 4 shows budgeted validity curves. EC differs from FullObs in that success is existential over completions of the unknown atoms: a formula is valid if, for each input world independently, some completion makes the formula match the observed target labels. This makes raw validity a permissive but meaningful feasibility criterion, while Acc@gold+25 asks whether the system can find a succinct completion-consistent explanation.

The core band is solvable by several systems. GPT-5.4 is strongest on EC-core (87.5% Acc@gold+25, 97.5% validity), ahead of GPT-5.2 (82.5%, 94.2%) and Opus 4.6 / Gemini 3.1 (81.7% Acc@gold+25). The hard band creates a much sharper test: GPT-5.4 again leads (28.7% Acc@gold+25, 87.5% validity), while GPT-5.2 is second

*Table 4.* **Within-problem bloat control (FullObs).** For problems with $\geq 2$ instance-correct predictions (excluding exact gold matches) across models, we compare held-out generalization of shortest vs longest formulas (and compact vs bloated when both exist). This controls for instance difficulty. $\Delta$ = short/compact − long/bloated; positive values indicate shorter formulas generalize better on the *same* problem. Fraction $\Delta < 0$: Short–Long 6%, Compact–Bloat 3%.

| Comparison | | Held-out Gen | | | | |
| | $n$ | Short | Long | $\Delta$ [CI] | $\Delta > 0$ | $p$ |
|---|---|---|---|---|---|---|
| Short–Long | 140 | 73.4% | 37.0% | +36.4 [29, 44] | 46% | <0.001 |
| Compact–Bloat | 61 | 87.1% | 11.8% | +75.3 [66, 84] | 92% | <0.001 |

Fixed-effects regression: $\beta_{AST\Delta} = -0.0063$ (SE=0.0003, $p = 0.000$), controlling for model and problem.

*Table 5.* **CI v1 failure mode decomposition.** YES-fail = formula does not match a YES world; NO-fail = formula exactly matches a contrastive NO target; Parse = output returned but formula extraction failed.

| Model | Correct | YES-fail | NO-fail | Parse | Missing |
|---|---|---|---|---|---|
| GPT-5.2 | **82.5%** | **17.0%** | **0.5%** | **0.0%** | **0.0%** |
| Opus 4.6 | 79.5% | 19.5% | 1.0% | **0.0%** | **0.0%** |
| GPT-5.4 | 79.0% | 20.0% | 1.0% | **0.0%** | **0.0%** |
| Gemini 3.5f | 78.5% | 20.0% | 1.5% | **0.0%** | **0.0%** |
| Grok4 | 78.0% | 19.5% | 2.5% | **0.0%** | **0.0%** |
| Gemini 3.1 | 70.0% | 23.0% | 7.0% | **0.0%** | **0.0%** |
| DeepSeek4Pro | 66.5% | 21.0% | 11.0% | **0.0%** | 1.5% |
| Grok4.1f | 60.5% | 25.0% | 12.0% | 2.0% | **0.0%** |
| Gemini 3 | 55.0% | 31.0% | 14.0% | **0.0%** | **0.0%** |
| Grok4.3 | 46.0% | 34.0% | 20.0% | **0.0%** | **0.0%** |

*Table 6.* **Held-out generalization by formula complexity (CI).** For instance-correct formulas, we compare YES held-out exact-match rates between *near-gold* formulas (AST $\leq$ gold+1) and *above-gold* formulas (AST $>$ gold+1). $\Delta$ = near-gold − above-gold; positive values indicate near-gold formulas generalize better to new YES worlds.

| Model | #Valid | Near-Gold Gen% | Above-Gold Gen% | $\Delta$ |
|---|---|---|---|---|
| GPT-5.2 | **165** | 7.2% | 2.3% | +5.0% |
| Opus 4.6 | 159 | 11.0% | 4.7% | +6.3% |
| GPT-5.4 | 158 | 12.1% | 1.9% | +10.2% |
| Gemini 3.5f | 157 | 14.8% | 1.4% | +13.4% |
| Grok4 | 156 | 14.7% | 5.9% | +8.8% |
| Gemini 3.1 | 140 | 10.4% | 5.6% | +4.9% |
| DeepSeek4Pro | 133 | **16.7%** | 2.0% | +14.6% |
| Grok4.1f | 121 | 10.3% | 1.6% | +8.7% |
| Gemini 3 | 110 | 11.6% | **9.3%** | +2.3% |
| Grok4.3 | 92 | 8.3% | 2.8% | +5.6% |

in budgeted accuracy at 25.0% but has far lower validity (53.8%). Overall, GPT-5.4 reaches 93.5% validity and 64.0% Acc@gold+25, the strongest EC result here.

The main EC lesson, however, is the gap between feasibility and parsimony. Unlike the FullObs and CI comparisons, GPT-5.4's EC improvement over GPT-5.2 does not come from lower bloat: GPT-5.4 is more bloated on EC (29.5% vs. 18.5%). Thus GPT-5.4 improves completion validity and budgeted performance, but does not close the EC parsimony gap. Even the best EC system relies substantially on oversized completion-consistent formulas.

As an additional diagnostic, we compute for invalid EC predictions the minimum number of label mismatches achievable under any completion of the unknown atoms (Appendix B.8). For most systems, invalid formulas remain several mismatches from EC validity even under the best completion, suggesting structural rather than one-label failures. GPT-5.4 is a notable exception: all of its invalid predictions fall in the 1–2 mismatch bucket, with mean minimum mismatch 1.0, so its residual failures are usually near-valid rather than structurally far off.

**Structural error analysis.** The task-level results above are partly explained by the structure of the planted formulas. For representative systems analyzed in Appendix B.5, we group instances by gold-formula structure and inspect failed

*Table 7.* **EC v1 accuracy by band.** Core = QD=1 (120 problems); Hard = QD=2 (80 problems).

| Model | Core @+25 | Core Val | Hard @+25 | Hard Val |
|---|---|---|---|---|
| GPT-5.4 | **87.5%** | **97.5%** | **28.7%** | **87.5%** |
| GPT-5.2 | 82.5% | 94.2% | 25.0% | 53.8% |
| Opus 4.6 | 81.7% | 82.5% | 20.0% | 21.2% |
| Grok4 | 78.3% | 78.3% | 15.0% | 15.0% |
| Gemini 3 | 75.8% | 76.7% | 16.2% | 18.8% |
| Gemini 3.1 | 81.7% | 81.7% | 10.0% | 10.0% |
| Gemini 3.5f | 75.0% | 75.0% | 15.0% | 16.2% |
| DeepSeek4Pro | 73.3% | 76.7% | 16.2% | 18.8% |
| Grok4.1f | 63.3% | 63.3% | 7.5% | 7.5% |
| Grok4.3 | 61.7% | 61.7% | 3.8% | 3.8% |

or over-budget predictions. The clearest recurring difficulty is universal relational generalization, especially in EC. Pure existential formulas are much easier than mixed existential/universal formulas: in EC, GPT-5.2 drops from 93.2% to 25.3% Acc@gold+25, Gemini 3 from 92.0% to 16.5%, and Grok4 from 86.4% to 15.2%. In FullObs, the same split is difficult for GPT-5.2 and Gemini 3, while Grok4 remains stronger. A recurring prediction-level error is to replace a condition over all relevant neighbors with a local witness, collapsing universal relational constraints such as

$$\forall y \, (R(x,y) \to \exists z \, (S(y,z) \land P(z)))$$

into existential clauses such as

$$\exists y \, (R(x,y) \land P(y)).$$

Appendix B.5 gives the full structural breakdowns, including feature-specific slices and returned-formula analyses.

**Summary of findings.** Taken together, the results separate three notions of success. A system may satisfy the input instance, find a compact formula, and generalize to held-out worlds; these are distinct. FullObs stresses compact multi-world induction, CI tests whether systems avoid shortcut explanations under contrastive evidence, and EC separates completion feasibility from succinct explanation. Across all three regimes, the strongest signal is that compactness matters: near-gold formulas generalize substantially better than bloated formulas, and structural failures often take the form of replacing global relational conditions with simpler local witnesses.

## 6. Related Work

**Inductive logic programming and relational concept learning.** Learning logical definitions from examples has a long history in inductive logic programming (ILP), including classic systems for learning relational rules and Horn-clause theories from structured data (Muggleton, 1991; Muggleton & Raedt, 1994; Quinlan, 1990; Srinivasan, 2001; Cropper & Muggleton, 2020). INDUCTION shares with ILP the goal of learning symbolic relational concepts from examples, and many of our generation concerns—avoiding trivial hypotheses, constructing informative examples, and analyzing structural failure modes—mirror ILP practice.

The setting differs in emphasis: our benchmark uses fully extensional finite structures, asks for a single first-order formula over a fixed output language, and evaluates both prompted language/reasoning models and symbolic synthesis baselines under the same exact task semantics, rather than proposing a new ILP algorithm.

**Program synthesis and constraint-based learning.** Synthesizing programs or formulas from constraints is central in program synthesis (Solar-Lezama et al., 2006; Alur et al., 2013; Torlak & Bodik, 2014). Our tasks can be viewed as formula synthesis problems whose specifications are given by multiple finite relational structures and target extensions. Unlike many synthesis benchmarks, INDUCTION emphasizes first-order quantifiers, relational structure, and generalization across worlds. Our generator also uses a benchmark-level analogue of counterexample-guided refinement (Solar-Lezama, 2008): additional worlds are constructed to eliminate plausible distractor hypotheses, yielding instances with controlled version-space properties.

**Logical and relational reasoning benchmarks.** A large body of work evaluates reasoning in logic settings, including datasets for rule-based inference, theorem proving, and natural-language first-order reasoning (Tafjord et al., 2021; Clark et al., 2020; Polu & Sutskever, 2020; Han et al., 2024). Many such benchmarks test deduction from given rules, often mediated through natural language. The CI task is also related to discriminative concept-learning settings such as Bongard or Zendo-style puzzles, but with fixed relational semantics and solver-verifiable evaluation (Bongard, 1970; Heath, 2001). A closely related finite-world benchmark is ABD, which uses a similar solver-checkable relational interface but asks for default–exception repairs, whereas INDUCTION asks for definitions of extensionally labeled target concepts (Batzoglou, 2026). INDUCTION tests a complementary inductive capability: recovering a compact first-order definition from extensional relational evidence. The inputs are finite structures rather than natural-language stories, and the outputs are symbolic formulas checked by exact model evaluation.

**Mathematical reasoning and formal proof benchmarks.** Recent benchmarks evaluate mathematical reasoning in natural language and in formal systems. Natural-language datasets include GSM8K-style word problems and competition mathematics such as MATH (Cobbe et al., 2021; Hendrycks et al., 2021). Formal benchmarks test proof or tactic generation in systems such as Coq and HOL Light, as well as formalization and autoformalization in datasets such as CoqGym, HOList, miniF2F, and ProofNet (Yang & Deng, 2019; Bansal et al., 2019; Zheng et al., 2022; Azerbayev et al., 2023). PROOFGRID evaluates deductive and equational reasoning through machine-checkable proofs in deliberately minimal formal notation (Arkoudas & Batzoglou,

2026). These benchmarks stress deductive proof search, proof checking, or formal proof construction. INDUCTION targets a different capability: synthesizing compact first-order hypotheses from examples and testing whether those hypotheses generalize to new finite structures.

**Neuro-symbolic learning and large language models.** Neuro-symbolic work studies how neural systems can represent, manipulate, or learn symbolic structure (d'Avila Garcez & Lamb, 2023; Polu & Sutskever, 2020). Large language and reasoning models can emit syntactically valid logical expressions, but their robustness and generalization remain difficult to assess. Prior evaluations have found that models may rely on shortcuts, surface patterns, or excessively long outputs when unconstrained (Wei et al., 2022; Dziri et al., 2023). INDUCTION contributes a setting where correctness is exact, finite-model-checkable, and paired with bloat-aware metrics, making it possible to separate merely satisfying the input constraints from finding compact hypotheses that generalize.

## 7. Limitations

**Controlled scope.** INDUCTION v1 uses a deliberately small relational signature, with unary predicates $P, Q$, binary predicates $R, S$, and small finite domains. This restriction is central to the benchmark design: it keeps evaluation exact, generation controllable, and errors interpretable. The results are not a claim about general first-order reasoning. Richer vocabularies, constants, functions, types, arithmetic, background theories, and larger domains may expose different failure modes.

**Template-bounded targets and non-unique explanations.** Gold concepts come from a curated bounded target language, and worlds are constructed to separate them from plausible distractors. This improves diagnostic clarity and structural coverage, but it can also induce template bias. Moreover, many distinct formulas can agree with the target labels on the finite input worlds. The planted gold formula is therefore not a claim of global syntactic minimality modulo first-order equivalence, and gold-relative budgets do not measure whether a returned formula is uniquely simplest.

**Task semantics and generation artifacts.** The three regimes intentionally isolate different aspects of induction, but their semantics introduce artifacts. EC's existential-completion semantics is permissive by design: large disjunctions can sometimes exploit completion flexibility or case-split across worlds, which is why we report bloat-aware metrics. CI's trap construction makes NO worlds informative, but the survivor and rejection filters can shift the realized distribution of accepted instances.

**Baselines and interfaces.** Our symbolic baselines are bounded searches under fixed encodings and compute budgets, not upper bounds on symbolic performance. Conversely, prompted-model evaluation depends on the prompt, output syntax, parser, decoding settings, and provider behavior; we do not tune prompts per model or perform a systematic compactness-prompt ablation. Appendix B.6 reports a small paired symbol-renaming pilot; performance does not collapse under consistent renaming, but this is not a full interface-robustness study.

## 8. Conclusion

We introduced INDUCTION, a solver-verifiable benchmark for synthesizing first-order definitions from finite relational worlds. Each instance asks for a single formula that explains an extensionally specified target concept across multiple structures, and every candidate can be checked exactly by finite-model evaluation and SMT. The three regimes—FULLOBS, CI, and EC—share this common interface while isolating different sources of difficulty: multi-world concept recovery, contrastive negative evidence, and reasoning under missing information.

The experiments show that current systems have real but incomplete ability to perform this kind of symbolic induction. Strong prompted models and symbolic baselines solve substantial parts of the benchmark, especially easier contrastive and completion settings, but sharp difficulty gradients remain. The hardest cases expose structural weaknesses such as universal relational generalization, lifted $x$-relations, and the tendency to replace global conditions with local existential witnesses.

The central empirical lesson is that validity is not enough. Some systems satisfy the input constraints with large case-splitting formulas, especially in settings where the finite worlds leave room for accidental shortcuts. Held-out evaluation shows that this distinction is consequential: among instance-correct solutions, compact near-gold formulas generalize much better than bloated ones. Parsimony is therefore not merely a preference for elegance; in this setting, it is a measurable signal of whether a formula has captured a stable concept.

More broadly, INDUCTION illustrates how to build logic benchmarks around exact semantics, controlled generation, diagnostics, and held-out tests of hypothesis stability. We hope it encourages evaluation protocols that reward not only consistency, but succinct explanations that remain correct under new evidence.

## Software and Data

The INDUCTION code, data, and experiment artifacts are available at https://github.com/SerafimBatzoglou/concept-synth. The release includes the v1 datasets, generation and evaluation code, analysis scripts, cached model outputs, structural-analysis artifacts, and the symbolic baseline code and results.

## Acknowledgements

This work used AI assistants, including Anthropic Claude Code and OpenAI Codex, for software development, drafting, and editorial improvements to manuscript and LaTeX scripts. All substantive scientific decisions, experiment design and execution, result verification, and final writing were performed and validated by the author. The author remains fully responsible for the content of this paper.

## Impact Statement

This work introduces benchmarks for inductive synthesis of first-order definitions and metrics that reward succinct, generalizable hypotheses. As with other benchmarks, it may bias research toward the covered distributions or incentivize gaming; we mitigate this by releasing diagnostics and recommending held-out splits or periodic regeneration.

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

## A. Symbolic Baselines

We added symbolic synthesis baselines to calibrate benchmark hardness and provide search-based symbolic reference points. These baselines are not intended to be an exhaustive survey of symbolic systems; rather, they cover three relevant comparisons: a unified SMT formula-synthesis baseline, a stronger FullObs-specific Z3 portfolio, and a restricted ILP/ASP-style comparator.

**Run protocol.** All symbolic rows use denominator=*all* benchmark instances for the corresponding task. Timeout or no-solution outcomes count as incorrect. The per-instance wall-clock cap is 30 minutes; individual component stages may use smaller internal budgets and may terminate earlier. Every returned symbolic hypothesis is decoded when necessary and rechecked by the same benchmark evaluator used for model outputs under the exact task semantics. For EC, this recheck uses the same world-local existential-completion semantics: in each world, the unknown ground atoms may be completed independently, but one completion must make the candidate formula match the whole target extension in that world.

**z3-prenex + rescue.** `z3-prenex` is the fair unified SMT baseline across FullObs, CI, and EC. It synthesizes prenex formulas using all prefixes over $\exists, \forall$ up to length two, followed by a bounded Boolean matrix over literals using the benchmark predicates $P, Q, R, S$ and variables $x, y, z$. The matrix is represented as a binary tree with leaf, conjunction, and disjunction nodes; leaves range over unary atoms on in-scope variables and binary atoms on ordered in-scope pairs, with negated atoms included directly. Candidate atoms with identical truth vectors over the finite grounding are deduplicated before search. The reported base runs search matrix depths three and four with a task-agnostic schedule over prefixes. For FullObs and EC, Z3 constrains the synthesized formula to match $T$ exactly for every object in every world. For CI, it constrains exact matching on YES worlds and requires at least one target-label mismatch on each NO world, matching the contrastive semantics. For EC, the encoding creates Z3 Boolean variables for unknown ground atoms, shared across all object constraints within a world, implementing the benchmark's per-world existential-completion semantics. The FullObs entry in Table 2 includes a rescue pass on unsolved cases, restricted to depth-three matrix searches with the QD=2 prefixes $\exists\exists$ and $\forall\exists$; this rescue adds 57 FullObs solutions beyond the base `z3-prenex` run. The CI and EC entries are the base unified runs.

**z3-ad-mix.** `z3-ad-mix` is a stronger FullObs-specific Z3 portfolio and is therefore not the fair unified baseline. Its first stage runs a FullObs-only portfolio that tries a small QD=1 template fallback and then compact schema encodings for common FullObs QD=2 families: universal-to-existential implications, guarded existential-universal patterns, and two-hop existential chains, with schema slots ranging over allowed literals built from $P, Q, R, S$. If that stage fails, `z3-ad-mix` runs a generic depth-4 QD=1 template search and then a generic depth-4 QD=2 template search. Its advantage comes from encoding dominant FullObs structural families as much smaller SMT problems than unrestricted formula search.

**ilasp-frag.** `ilasp-frag` is a restricted ILP/ASP-style comparator for FullObs and CI; EC is not supported by this encoding. ILASP learns answer-set programs from examples subject to a *mode bias*, which specifies the predicates, variables, and rule shapes allowed in the search. We encode each finite world as a context containing facts for `obj/1`, `p/1`, `q/1`, `r/2`, and `s/2`, and ask ILASP to learn a nonrecursive target predicate `t/1`. In the reported runs, the mode bias uses at most four variables and a maximum penalty of 32; there is no predicate invention and no recursive use of `t` in rule bodies.

The learned rule set is decoded back into a benchmark first-order formula and rechecked by the benchmark evaluator. Under this decoder, multiple rules for `t/1` become a disjunction of rule bodies, and variables not appearing in the head are existentially quantified. Thus the reported fragment corresponds to disjunctions of existential conjunctions of observed-predicate literals; it does not target the full benchmark output grammar and is substantially weaker than the Z3 search spaces. For FullObs, each world requires exact agreement with the $T$-true and $T$-false objects. For CI, YES worlds require exact agreement, while NO worlds are encoded so that a rule set exactly matching a NO target is rejected.

**cvc5/SyGuS.** We also attempted a grammar-based SyGuS baseline using `cvc5`. We do not report it as a main row: the stable formulation was much weaker than the Z3 baselines, and a cleaner revised formulation was not robust enough for benchmark-scale runs. We include this note to document the attempted SyGuS route, but omit `cvc5` numbers from the symbolic-baseline table.

## B. Additional Tables and Figures

This appendix provides diagnostics behind the main results: where the benchmark becomes hard, when validity depends on formula bloat, which structural families remain difficult, and how failed formulas differ from compact successful ones.

*Table 8.* **FullObs v1 band-wise accuracy** (Acc_all, denominator = total problems per band). Extremes are aggregated.

| Model | simple | easy | medium | hard | extreme |
|---|---|---|---|---|---|
| GPT-5.4 | 96.0% | 71.0% | 39.0% | 25.0% | 8.0% |
| GPT-5.2 | **100.0%** | **89.0%** | 29.0% | 21.0% | 0.0% |
| Grok4 | **100.0%** | 75.0% | **43.0%** | **39.0%** | **16.0%** |
| Gemini 3.5f | 96.0% | 48.0% | 27.0% | 26.0% | 8.0% |
| Opus 4.6 | 92.0% | 43.0% | 11.0% | 10.0% | 4.0% |
| DeepSeek4Pro | 88.0% | 37.0% | 25.0% | 18.0% | 6.0% |
| Gemini 3.1 | 96.0% | 31.0% | 6.0% | 5.0% | 0.0% |
| Gemini 3 | 96.0% | 26.0% | 5.0% | 4.0% | 2.0% |
| Grok4.1f | 92.0% | 26.0% | 8.0% | 8.0% | 4.0% |
| Grok4.3 | 96.0% | 25.0% | 0.0% | 2.0% | 2.0% |
| DSR | 68.0% | 13.0% | 7.0% | 2.0% | 0.0% |
| Opus 4.5 | 80.0% | 10.0% | 1.0% | 1.0% | 0.0% |
| Hermes4 | 40.0% | 0.0% | 0.0% | 0.0% | 0.0% |
| GPT-4o | 0.0% | 0.0% | 0.0% | 0.0% | 0.0% |

## B.1. FullObs Diagnostic Tables

FullObs diagnostics support three conclusions. First, the band split confirms a steep difficulty gradient: several systems nearly saturate the simple QD=1 slice, but accuracy drops sharply on medium, hard, and extreme slices (Table 8). Second, raw accuracy and budgeted accuracy diverge for some strong models, especially GPT-5.2, indicating that many correct formulas are oversized (Table 9). Third, family and size breakdowns show that the difficulty is structural rather than uniform: some families are solved reliably by frontier systems, while others remain near zero, and valid predictions vary widely in compactness (Tables 10 and 11).

*Table 9.* **FullObs v1 overall accuracy** (sorted by Acc@gold+25). Bold = best per column.

| Model | @+25 | Acc_all | Cov | Bloat |
|---|---|---|---|---|
| Grok4 | **46.7%** | **50.7%** | 89.3% | 4.0% |
| Gemini 3.5f | 33.9% | 34.4% | 99.7% | 0.5% |
| GPT-5.4 | 32.0% | 43.5% | **100.0%** | 11.5% |
| DeepSeek4Pro | 25.3% | 28.0% | 99.2% | 2.7% |
| GPT-5.2 | 19.5% | 43.7% | **100.0%** | 24.3% |
| Opus 4.6 | 18.9% | 23.7% | **100.0%** | 4.8% |
| Grok4.1f | 17.9% | 17.9% | 98.4% | **0.0%** |
| Gemini 3.1 | 17.6% | 17.6% | **100.0%** | **0.0%** |
| Gemini 3 | 15.2% | 16.0% | **100.0%** | 0.8% |
| Grok4.3 | 13.9% | 13.9% | 91.2% | **0.0%** |
| DSR | 10.4% | 10.4% | 99.7% | **0.0%** |
| Opus 4.5 | 8.5% | 8.5% | 98.7% | **0.0%** |
| Hermes4 | 2.7% | 2.7% | 99.5% | **0.0%** |
| GPT-4o | 0.0% | 0.0% | **100.0%** | **0.0%** |

*Table 10.* **FullObs v1 Acc@gold+25 by formula family for selected prompted models.**

| Family | GPT-5.4 | GPT-5.2 | Grok4 | Gemini 3.5f | Opus 4.6 |
|---|---|---|---|---|---|
| A | 66.7% | 66.7% | **100.0%** | 66.7% | 55.6% |
| B | 13.6% | 13.6% | **15.3%** | 10.2% | 10.2% |
| C | 5.8% | 7.7% | **11.5%** | 5.8% | 7.7% |
| D | 39.6% | 8.3% | **81.2%** | 64.6% | 12.5% |
| F | 31.4% | 2.9% | **65.7%** | 51.4% | 5.7% |
| G | 0.0% | 0.0% | **9.1%** | 0.0% | **9.1%** |
| H | 58.8% | 27.1% | **67.1%** | 42.4% | 24.7% |
| M | 0.0% | 0.0% | **28.6%** | 9.5% | 14.3% |
| oth | 54.8% | **64.3%** | 59.5% | 59.5% | 54.8% |
| Z | 0.0% | 0.0% | 0.0% | 0.0% | 0.0% |

*Table 11.* **FullObs v1 formula size breakdown for successful predictions.** Compact = AST < gold; Equal = gold ≤ AST ≤ gold+1; Longer = gold+1 < AST ≤ gold+25; Bloat = AST > gold+25.

| Model | Compact | Equal | Longer | Bloat |
|---|---|---|---|---|
| Grok4 | 2.7% | **38.1%** | 5.9% | 4.0% |
| Gemini 3.5f | 3.2% | 27.7% | 2.9% | 0.5% |
| GPT-5.4 | 3.2% | 22.1% | 6.7% | 11.5% |
| DeepSeek4Pro | 2.4% | 18.7% | 4.3% | 2.7% |
| GPT-5.2 | 2.4% | 9.6% | 7.5% | 24.3% |
| Opus 4.6 | 2.7% | 12.0% | 4.3% | 4.8% |
| Grok4.1f | 2.4% | 13.9% | 1.6% | **0.0%** |
| Gemini 3.1 | **3.5%** | 13.3% | 0.8% | **0.0%** |
| Gemini 3 | 3.2% | 6.4% | 5.6% | 0.8% |
| Grok4.3 | 2.4% | 10.1% | 1.3% | **0.0%** |
| DSR | 1.9% | 6.1% | 2.4% | **0.0%** |
| Opus 4.5 | 2.7% | 4.0% | 1.9% | **0.0%** |
| Hermes4 | 0.8% | 1.9% | **0.0%** | **0.0%** |
| GPT-4o | 0.0% | 0.0% | **0.0%** | **0.0%** |

## B.2. CI Diagnostic Tables

CI diagnostics distinguish contrastive success from recovery of the planted concept. Overall CI scores are high for frontier systems, but this does not mean the planted concept is uniquely identified: many formulas can satisfy the YES/NO criterion without matching $\varphi^\star$ on new worlds. We therefore analyze performance by family and decompose failures into YES-failures and NO-failures. The overall and family tables show which structural families remain difficult (Tables 12 and 13), while the band-specific failure profiles show whether systems fail by missing YES worlds or by falling into shortcut hypotheses that exactly match NO targets (Tables 14 and 15).

*Table 12.* **CI v1 overall accuracy** (sorted by Acc@gold+25). Bold = best per column.

| Model | @+25 | Acc_all | Cov | Bloat |
|---|---|---|---|---|
| Gemini 3.5f | **78.0%** | 78.5% | **100.0%** | 0.5% |
| GPT-5.4 | 76.0% | 79.0% | **100.0%** | 3.0% |
| Grok4 | 75.5% | 78.0% | **100.0%** | 2.5% |
| Opus 4.6 | 75.5% | 79.5% | **100.0%** | 4.0% |
| GPT-5.2 | 73.0% | **82.5%** | **100.0%** | 9.5% |
| Gemini 3.1 | 70.0% | 70.0% | **100.0%** | **0.0%** |
| DeepSeek4Pro | 63.5% | 66.5% | 98.5% | 3.0% |
| Grok4.1f | 60.5% | 60.5% | 98.0% | **0.0%** |
| Gemini 3 | 55.0% | 55.0% | **100.0%** | **0.0%** |
| Grok4.3 | 45.0% | 46.0% | **100.0%** | 1.0% |
| DSR | 41.5% | 41.5% | 99.0% | **0.0%** |
| Opus 4.5 | 34.0% | 34.0% | **100.0%** | **0.0%** |
| Hermes4 | 2.5% | 2.5% | 99.5% | **0.0%** |
| GPT-4o | 0.5% | 0.5% | 99.0% | **0.0%** |

*Table 13.* **CI v1 Acc@gold+25 by formula family for selected prompted models.**

| Family | GPT-5.4 | GPT-5.2 | Grok4 | Gemini 3.5f | Opus 4.6 |
|---|---|---|---|---|---|
| A | **100.0%** | **100.0%** | **100.0%** | **100.0%** | **100.0%** |
| B | 31.8% | 27.3% | **36.4%** | 27.3% | **36.4%** |
| C | **10.5%** | **10.5%** | 5.3% | **10.5%** | 5.3% |
| D | 75.0% | 80.0% | 70.0% | 80.0% | **90.0%** |
| F | 84.2% | 89.5% | 89.5% | **100.0%** | 89.5% |
| G | **100.0%** | **100.0%** | **100.0%** | **100.0%** | **100.0%** |
| H | **92.3%** | 84.6% | **92.3%** | **92.3%** | **92.3%** |
| oth | 90.0% | **100.0%** | **100.0%** | 90.0% | 80.0% |
| Z | 93.5% | 85.5% | 90.3% | **95.2%** | 87.1% |

*Table 14.* **CI v1 failure modes: core band.** YES-fail = formula does not match a YES world; NO-fail = formula exactly matches a contrastive NO target.

| Model | Correct | YES-fail | NO-fail |
|---|---|---|---|
| GPT-5.2 | **85.0%** | **14.2%** | **0.8%** |
| GPT-5.4 | 81.7% | 16.7% | 1.7% |
| Gemini 3.5f | 81.7% | 17.5% | **0.8%** |
| Opus 4.6 | 80.8% | 17.5% | 1.7% |
| Grok4 | 80.0% | 16.7% | 3.3% |
| Gemini 3.1 | 70.8% | 21.7% | 7.5% |
| DeepSeek4Pro | 65.8% | 18.3% | 15.0% |
| Grok4.1f | 61.7% | 20.8% | 14.2% |
| Gemini 3 | 58.3% | 27.5% | 14.2% |
| DSR | 41.7% | 43.3% | 15.0% |
| Grok4.3 | 40.0% | 35.0% | 25.0% |
| Opus 4.5 | 34.2% | 37.5% | 28.3% |
| Hermes4 | 3.3% | 85.8% | 10.8% |
| GPT-4o | 0.8% | 95.0% | 3.3% |

*Table 15.* **CI v1 failure modes: lift_mix band.** YES-fail = formula does not match a YES world; NO-fail = formula exactly matches a contrastive NO target.

| Model | Correct | YES-fail | NO-fail |
|---|---|---|---|
| GPT-5.2 | **78.8%** | **21.2%** | **0.0%** |
| Opus 4.6 | 77.5% | 22.5% | **0.0%** |
| GPT-5.4 | 75.0% | 25.0% | **0.0%** |
| Grok4 | 75.0% | 23.8% | 1.2% |
| Gemini 3.5f | 73.8% | 23.8% | 2.5% |
| Gemini 3.1 | 68.8% | 25.0% | 6.2% |
| DeepSeek4Pro | 67.5% | 25.0% | 5.0% |
| Grok4.1f | 58.8% | 31.2% | 8.8% |
| Grok4.3 | 55.0% | 32.5% | 12.5% |
| Gemini 3 | 50.0% | 36.2% | 13.8% |
| DSR | 41.2% | 42.5% | 13.8% |
| Opus 4.5 | 33.8% | 41.2% | 25.0% |
| Hermes4 | 1.2% | 81.2% | 16.2% |
| GPT-4o | 0.0% | 88.8% | 10.0% |

## B.3. EC Diagnostic Tables

EC diagnostics distinguish completion feasibility from succinct explanation. Because EC validity is existential over completions, a formula can be valid while still being much larger than the planted concept. We therefore report both unbounded validity and Acc@gold+25, then inspect family structure and size breakdowns. The overall table highlights the validity/budgeted-accuracy gap, especially for GPT-5.4 and GPT-5.2 (Table 16); the family table identifies which formula families remain difficult under completion semantics (Table 17); and the size breakdown shows which systems obtain validity with compact formulas versus bloated ones (Table 18).

## B.4. Error Profiles

Beyond binary success, we analyze the direction and margin of errors. For FullObs, false positives and false negatives reveal whether a system's formula is too permissive or too restrictive. For CI, YES-world FP/FN rates measure ordinary fit to YES worlds, while the NO margin measures separation from contrastive NO targets: a NO failure is exactly the zero-margin case where the formula matches a NO target. The pattern helps explain failure modes (Table 19): strong FullObs systems tend to reduce FP and FN simultaneously, whereas weaker systems often show high FP, high FN, or both. In CI, NO margin complements the YES/NO-failure decomposition by measuring how far nonmatching formulas are from the forbidden NO targets.

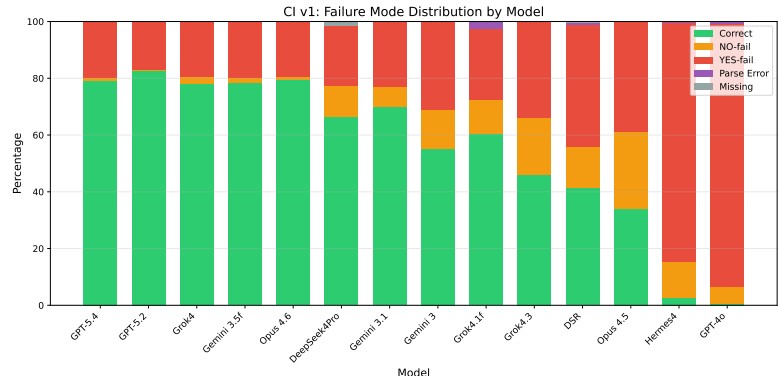

*Figure 3.* CI v1 failure mode distribution by model (stacked bar chart).

*Table 16.* **EC v1 overall accuracy** (sorted by Acc@gold+25). Bold = best per column.

| Model | @+25 | Valid | Cov | Bloat |
|---|---|---|---|---|
| GPT-5.4 | **64.0%** | **93.5%** | 99.5% | 29.5% |
| GPT-5.2 | 59.5% | 78.0% | **100.0%** | 18.5% |
| Opus 4.6 | 57.0% | 58.0% | 99.5% | 1.0% |
| Grok4 | 53.0% | 53.0% | 99.5% | **0.0%** |
| Gemini 3.1 | 53.0% | 53.0% | **100.0%** | **0.0%** |
| Gemini 3 | 52.0% | 53.5% | **100.0%** | 1.5% |
| Gemini 3.5f | 51.0% | 51.5% | **100.0%** | 0.5% |
| DeepSeek4Pro | 50.5% | 53.5% | 99.5% | 3.0% |
| Grok4.1f | 41.0% | 41.0% | 98.5% | **0.0%** |
| Grok4.3 | 38.5% | 38.5% | **100.0%** | **0.0%** |
| DSR | 33.0% | 33.0% | **100.0%** | **0.0%** |
| Opus 4.5 | 30.0% | 30.0% | 99.0% | **0.0%** |
| Hermes4 | 15.5% | 15.5% | **100.0%** | **0.0%** |
| GPT-4o | 2.0% | 2.0% | **100.0%** | **0.0%** |

*Table 17.* **EC v1 Acc@gold+25 by formula family for selected prompted models.**

| Family | GPT-5.4 | GPT-5.2 | Grok4 | Gemini 3.5f | Opus 4.6 |
|---|---|---|---|---|---|
| A | **28.6%** | 0.0% | 16.7% | 14.3% | 14.3% |
| B | 58.8% | **67.6%** | 44.1% | 47.1% | 61.8% |
| C | **65.0%** | 60.0% | 55.0% | 60.0% | 50.0% |
| D | **16.0%** | 7.7% | 7.7% | 3.8% | 7.7% |
| F | 7.7% | **15.4%** | 7.7% | 7.7% | 7.7% |
| G | **66.7%** | 0.0% | 33.3% | 33.3% | 33.3% |
| oth | **88.7%** | 82.5% | 77.3% | 72.2% | 80.4% |

*Table 18.* **EC v1 formula size breakdown for valid predictions.** Compact = AST < gold; Equal = gold ≤ AST ≤ gold+1; Longer = gold+1 < AST ≤ gold+25; Bloat = AST > gold+25.

| Model | Compact | Equal | Longer | Bloat |
|---|---|---|---|---|
| GPT-5.4 | 24.5% | 15.0% | 24.5% | 29.5% |
| GPT-5.2 | 21.0% | 19.5% | 19.0% | 18.5% |
| Opus 4.6 | 25.0% | 15.5% | 16.5% | 1.0% |
| Grok4 | 29.0% | **20.5%** | 3.5% | **0.0%** |
| Gemini 3.1 | **32.0%** | 19.5% | 1.5% | **0.0%** |
| Gemini 3 | 22.0% | 20.0% | 10.0% | 1.5% |
| Gemini 3.5f | 29.0% | 16.5% | 5.5% | 0.5% |
| DeepSeek4Pro | 20.0% | 17.0% | 13.5% | 3.0% |
| Grok4.1f | 22.5% | 15.5% | 3.0% | **0.0%** |
| Grok4.3 | 24.5% | 11.5% | 2.5% | **0.0%** |
| DSR | 20.0% | 10.5% | 2.5% | **0.0%** |
| Opus 4.5 | 16.5% | 7.0% | 6.5% | **0.0%** |
| Hermes4 | 11.5% | 4.0% | **0.0%** | **0.0%** |
| GPT-4o | 1.5% | 0.5% | **0.0%** | **0.0%** |

*Table 19.* **Input-world error profiles.** Mean per-world false positive (FP%) and false negative (FN%) rates. For FullObs, rates are averaged across input worlds. For CI, YES FP/FN are computed on YES worlds only; NO Margin is the mean number of mismatched elements on NO worlds, where higher values indicate stronger separation from NO targets.

| Model | FullObs | | CI | | |
| | FP% | FN% | YES FP% | YES FN% | NO Marg |
|---|---|---|---|---|---|
| Grok4 | **11.2** | **4.3** | 5.2 | 2.9 | 2.8 |
| GPT-5.2 | 11.8 | 6.4 | 3.6 | 2.5 | 3.2 |
| GPT-5.4 | 12.4 | 6.2 | **3.6** | **1.6** | 2.9 |
| Gemini 3.5f | 13.3 | 11.2 | 4.1 | 3.6 | 3.2 |
| DeepSeek4Pro | 14.3 | 12.3 | 3.9 | 4.3 | 2.7 |
| Opus 4.6 | 13.7 | 9.6 | 4.1 | 2.5 | 2.8 |
| Grok4.1f | 16.4 | 12.1 | 5.1 | 5.0 | 2.7 |
| Gemini 3.1 | 17.9 | 10.5 | 5.2 | 2.8 | 2.7 |
| Gemini 3 | 17.8 | 10.4 | 5.8 | 4.0 | 2.7 |
| Grok4.3 | 21.0 | 8.6 | 6.4 | 3.8 | 2.7 |
| DSR | 19.0 | 16.3 | 6.7 | 7.8 | 2.8 |
| Opus 4.5 | 20.1 | 13.8 | 6.5 | 5.3 | 2.8 |
| Hermes4 | 27.7 | 16.5 | 14.2 | 16.4 | 3.7 |
| GPT-4o | 38.2 | 7.4 | 24.2 | 11.9 | **3.8** |

## B.5. Structural Error Analysis

This section analyzes which logical structures current systems miss. We group instances by features of the planted gold formula and separately inspect failed or over-budget predictions. The main finding is that difficulty is not merely a function of formula size or quantifier count: EC shows a particularly sharp drop on universal relational structure, and failed predictions often replace global universal conditions with local existential witnesses. The release repository includes the scripts and intermediate CSV files used to generate these analyses. The tables below report representative slices for three prompted models.

*Table 20.* Acc@gold+25 by existential vs. mixed existential/universal structure. Rows include the number of instances in each slice.

| Slice | GPT-5.2 | Gemini 3 | Grok4 |
|---|---|---|---|
| FullObs pure existential ($n = 107$) | 41.1% | 33.6% | 73.8% |
| FullObs mixed existential/universal ($n = 265$) | 9.8% | 7.2% | 35.1% |
| EC pure existential ($n = 88$) | 93.2% | 92.0% | 86.4% |
| EC mixed existential/universal ($n = 79$) | 25.3% | 16.5% | 15.2% |

*Table 21.* Effect of placing $x$ only in the premise of a universal rule. Each entry compares Acc@gold+25 on formulas without this feature versus formulas where the free variable $x$ appears only in the premise/guard of a universal rule. EC shows a consistent drop; FullObs is more model-dependent.

| Slice | GPT-5.2 | Gemini 3 | Grok4 |
|---|---|---|---|
| FullObs ($n = 280 \rightarrow 95$) | $21.1\% \rightarrow 14.7\%$ | $16.4\% \rightarrow 11.6\%$ | $36.1\% \rightarrow 77.9\%$ |
| EC ($n = 138 \rightarrow 62$) | $76.8\% \rightarrow 21.0\%$ | $71.0\% \rightarrow 9.7\%$ | $63.8\% \rightarrow 29.0\%$ |

We call a formula helper-like when it effectively inlines an auxiliary relational condition by introducing a witness and then imposing a second relational condition through that witness. Operationally, the analysis uses the generator's subfamily metadata to identify QD=2 gold formulas with nontrivial inner relational structure.

*Table 22.* Effect of inlined helper-like relational structure. Each entry compares Acc@gold+25 without versus with the feature. The effect is strongest and most consistent on EC; FullObs effects are model-dependent.

| Slice | GPT-5.2 | Gemini 3 | Grok4 |
|---|---|---|---|
| FullObs ($n = 131 \rightarrow 244$) | $29.8\% \rightarrow 13.9\%$ | $26.0\% \rightarrow 9.4\%$ | $28.2\% \rightarrow 56.6\%$ |
| EC ($n = 155 \rightarrow 45$) | $74.2\% \rightarrow 8.9\%$ | $66.5\% \rightarrow 2.2\%$ | $65.8\% \rightarrow 8.9\%$ |

**Universal-to-existential collapse.** The clearest returned-formula pattern is universal-to-existential collapse. For EC gold formulas with universal structure, failed or over-budget predictions are often pure existential. This suggests a structural simplification error: a condition over all relational successors of $x$ is replaced by the existence of one local witness. For example, a gold pattern such as

$$\forall y \, (R(x,y) \rightarrow \exists z \, (S(y,z) \wedge P(z)))$$

is often replaced by a local existential clause such as

$$\exists y \, (R(x,y) \wedge P(y)).$$

*Table 23.* Universal-to-existential collapse in EC. Entries report the fraction of failed or over-budget predictions whose returned formula is pure existential, restricted to the indicated universal gold slice. Parentheses give the number of failed or over-budget predictions in the denominator.

| Model | Universal golds | $x$ only in universal premise |
|---|---|---|
| GPT-5.2 | 81.1% ($n = 74$) | 83.7% ($n = 49$) |
| Gemini 3 | 73.9% ($n = 88$) | 67.9% ($n = 56$) |
| Grok4 | 70.4% ($n = 81$) | 72.7% ($n = 44$) |

Models fail differently on these slices. On the hard nested EC slice, defined here by the helper-like inlined tag ($n = 45$), GPT-5.2 reaches 42.2% raw validity but only 8.9% Acc@gold+25, indicating many valid-but-bloated surrogates; Gemini 3 is at 4.4% validity and 2.2% Acc@gold+25, and Grok4 at 8.9% and 8.9%, indicating more outright failure.

**Worked example.** One GPT-5.2 EC failure from instance EC hard 0020 has gold formula

$$\varphi^\star(x) = \forall y \left( S(x, y) \to \exists z \left( S(y, z) \land Q(z) \right) \right).$$

The returned formula is invalid, 46 AST nodes larger than the gold formula, and existential-only:

$$\begin{aligned}
\hat{\varphi}(x) =& (\neg P(x) \land \exists y \left( S(x, y) \land Q(y) \right)) \\
& \lor (\neg P(x) \land \exists y \left( R(x, y) \land P(y) \land Q(y) \right) \land \exists z \left( R(x, z) \land P(z) \land \neg Q(z) \right)) \\
& \lor (P(x) \land Q(x) \land R(x, x) \land \exists y \left( S(x, y) \land \neg Q(y) \right)).
\end{aligned}$$

The prediction replaces a universal condition over all $S$-successors of $x$ with a finite disjunction of local existential witnesses, illustrating the collapse measured in Table 23.

### B.6. Symbol-Renaming Pilot

To probe sensitivity to predicate and object names, we ran a small paired symbol-renaming pilot on 30 instances: 10 FullObs, 10 CI, and 10 EC. We consistently renamed all symbols throughout the task text and instance data: $P \to \mathrm{Foo}$, $Q \to \mathrm{Bar}$, $R \to \mathrm{Blorp}$, $S \to \mathrm{Wump}$, and $a_i \to \mathrm{dax}_i$. On this paired sample, overall accuracy changed from 40.0% to 33.3% for Gemini 3 and from 66.7% to 73.3% for GPT-5.2, with 100% parse rate throughout. This pilot does not establish full symbol invariance, but it suggests that performance does not collapse under consistent renaming of the abstract predicates and objects.

*Table 24.* Symbol-renaming pilot on 30 paired instances: 10 FullObs, 10 CI, and 10 EC.

| Model | Original | Renamed | Parse |
|---|---|---|---|
| Gemini 3 | 40.0% | 33.3% | 100.0% |
| GPT-5.2 | 66.7% | 73.3% | 100.0% |

### B.7. EC Budget Curves

The EC budget curves visualize the gap between completion feasibility and succinct explanation. Systems such as GPT-5.4 and GPT-5.2 continue to gain solved instances as the AST budget increases, showing that some EC validity relies on formulas well above the planted gold size (Figure 4).

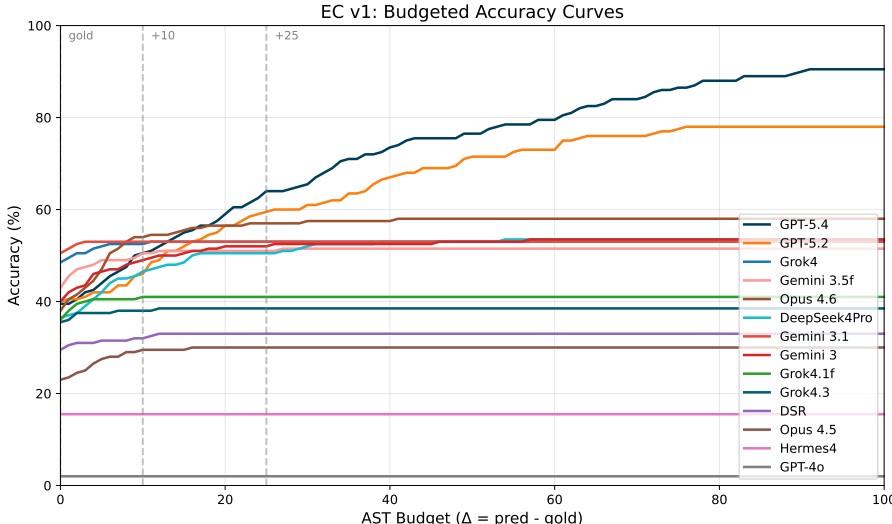

*Figure 4.* EC v1 budgeted validity curves: fraction of instances solved by formulas satisfying EC semantics and $\mathrm{AST}(\hat{\varphi}) \le \mathrm{AST}(\varphi^\star) + \Delta$.

### B.8. EC Best-Completion Error Analysis

For EC predictions that fail validity, we ask whether the failure is close to repairable by a completion of the unknown atoms. For each invalid formula, we compute the minimum number of target-label mismatches achievable under any completion. A minimum mismatch of 0 would imply EC validity; larger values indicate distance from feasibility. The diagnostic shows that most invalid predictions are not single-label mistakes: for most systems, invalid formulas remain at least three mismatches

away even under the best completion (Table 25). GPT-5.4 is the main exception: all of its invalid EC predictions fall in the 1–2 mismatch bucket, with mean minimum mismatch 1.0, suggesting that its residual EC failures are usually near-valid rather than structurally far off. This diagnostic is not a replacement for budgeted accuracy; it distinguishes near-valid hypotheses from formulas that cannot plausibly be repaired by completion alone.

*Table 25.* **EC best-completion error analysis.** For predictions that fail the existential-completion (EC) validity check, we compute the *minimum mismatches* achievable under any completion of unknown atoms. A formula with min-mismatch=0 would be EC-valid; higher values indicate how far the formula is from validity. Mean MM = mean minimum mismatches for invalid predictions. Distribution columns show the fraction of invalid predictions in each mismatch range.

| Model | Total | Valid | Mean MM | 1–2 | ≥3 |
|---|---|---|---|---|---|
| GPT-5.4 | 135 | **130** (96%) | **1.0** | **100%** | **0%** |
| GPT-5.2 | 137 | 120 (88%) | 5.6 | 35% | 65% |
| Opus 4.6 | 136 | 102 (75%) | 4.2 | 38% | 62% |
| Gemini 3.1 | 136 | 100 (74%) | 5.4 | 11% | 89% |
| DeepSeek4Pro | 136 | 99 (73%) | 3.9 | 38% | 62% |
| Grok4 | 136 | 96 (71%) | 4.7 | 32% | 68% |
| Gemini 3 | 137 | 95 (69%) | 3.6 | 38% | 62% |
| Gemini 3.5f | 136 | 94 (69%) | 4.2 | 37% | 63% |
| Grok4.1f | 137 | 76 (55%) | 4.8 | 29% | 71% |
| Grok4.3 | 136 | 74 (54%) | 5.2 | 23% | 77% |
| DSR | 137 | 63 (46%) | 5.0 | 19% | 81% |
| Opus 4.5 | 137 | 57 (42%) | 5.7 | 13% | 87% |
| Hermes4 | 137 | 28 (20%) | 5.5 | 20% | 80% |
| GPT-4o | 137 | 3 (2%) | 7.9 | 4% | 96% |

## B.9. CI Generalization vs Bloat

CI held-out generalization is low in absolute terms because CI correctness does not uniquely identify the planted gold concept. The relevant question is therefore comparative: among instance-correct CI formulas, do shorter formulas generalize better than longer ones? The AST-bin curve and within-problem control both answer yes, though the effect is smaller than in FullObs (Figure 5; Table 26).

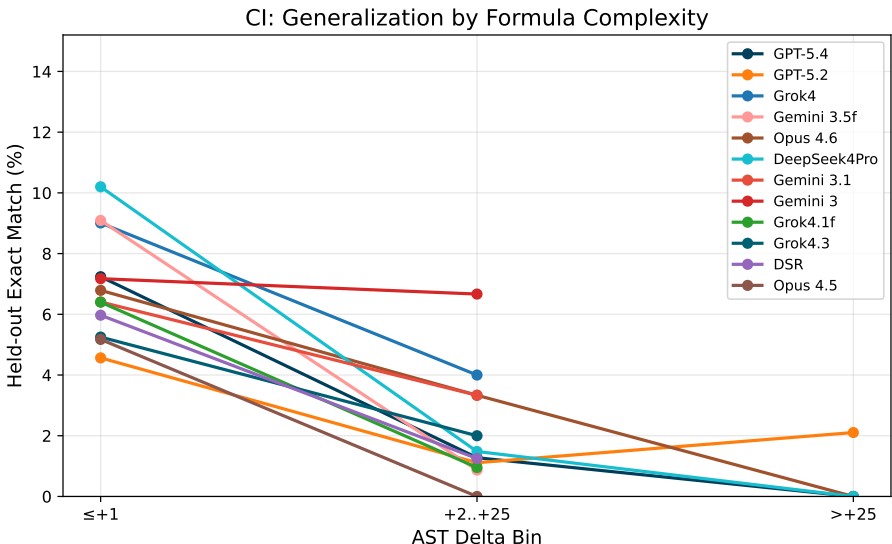

*Figure 5.* CI held-out generalization by AST delta bin. Same methodology as FullObs; model-bin points with at most five predictions are omitted. Bloated formulas generalize worse to new YES worlds. Curves show selected prompted models; full numerical results are in the tables.

## B.10. Lift-Hard Breakdown

Lift-hard formulas are a targeted stress test for universal relational generalization: relations involving the free variable $x$ appear inside universally quantified contexts, making it easy for systems to drop the lifted literal or confuse argument

*Table 26.* **Within-problem bloat control (CI).** For problems with $\geq 2$ instance-correct predictions (excluding exact gold matches) across models, we compare held-out generalization of shortest vs longest formulas (and compact vs bloated when both exist). This controls for instance difficulty. $\Delta$ = short/compact $-$ long/bloated; positive values indicate shorter formulas generalize better on the *same* problem. Fraction $\Delta < 0$: Short–Long 2%, Compact–Bloat 5%.

| | | Held-out Gen | | | |
|---|---|---|---|---|---|
| Comparison | $n$ | Short | Long | $\Delta$ [CI] | $\Delta > 0$ $p$ |
| Short–Long | 167 | 6.2% | 3.0% | +3.2 [1, 5] | 8% 0.02 |
| Compact–Bloat | 102 | 6.4% | 1.7% | +4.7 [2, 8] | 14% 0.03 |

Fixed-effects regression: $\beta_{\text{AST}\Delta} = -0.0024$ (SE=0.0002, $p = 0.000$), controlling for model and problem.

*Table 27.* **Lift-hard breakdown across tasks.** Lift-hard instances contain cross-relational patterns (using both R and S predicates) that empirically are harder for most systems. We report Acc@gold+25 (budgeted accuracy) separately for lift-hard and non-lift instances. EC v1 contains no lift-hard instances by construction (N/A).

| | FullObs | | CI | | EC | |
|---|---|---|---|---|---|---|
| Model | Lift (61) | Non (314) | Lift (28) | Non (172) | Lift (0) | Non (200) |
| GPT-5.4 | 0.0% | 38.2% | 53.6% | 79.7% | N/A | 64.0% |
| GPT-5.2 | 0.0% | 23.2% | 50.0% | 76.7% | N/A | 59.5% |
| Grok4 | 13.1% | 53.2% | 71.4% | 76.2% | N/A | 53.0% |
| Gemini 3.5f | 1.6% | 40.1% | 64.3% | 80.2% | N/A | 51.0% |
| Opus 4.6 | 4.9% | 21.7% | 60.7% | 77.9% | N/A | 57.0% |
| DeepSeek4Pro | 0.0% | 30.3% | 50.0% | 65.7% | N/A | 50.5% |
| Gemini 3.1 | 0.0% | 21.0% | 67.9% | 70.3% | N/A | 53.0% |
| Gemini 3 | 0.0% | 18.2% | 46.4% | 56.4% | N/A | 52.0% |
| Grok4.1f | 0.0% | 21.3% | 46.4% | 62.8% | N/A | 41.0% |
| Grok4.3 | 0.0% | 16.6% | 53.6% | 43.6% | N/A | 38.5% |
| DSR | 0.0% | 12.4% | 46.4% | 40.7% | N/A | 33.0% |
| Opus 4.5 | 0.0% | 10.2% | 32.1% | 34.3% | N/A | 30.0% |
| Hermes4 | 0.0% | 3.2% | 0.0% | 2.9% | N/A | 15.5% |
| GPT-4o | 0.0% | 0.0% | 0.0% | 0.6% | N/A | 2.0% |

orientation. The breakdown confirms that lift-hard instances are substantially harder than non-lift instances in FullObs and CI (Table 27). EC v1 contains no lift-hard instances by construction; this avoids near-zero success in the partial-observation setting while EC nested-quantifier hardness is calibrated separately.

## B.11. Generation Design Lessons

Several methodological lessons emerged from generation experiments:

1. **Survivor bands matter.** In CI, the number of traps surviving after YES worlds strongly affects difficulty. Too many survivors make problems easy; too few make generation fail. The tight band $[2, 4]$ balances these.
2. **Frozen pools enable diagnostics.** Using a frozen hypothesis pool (rather than per-problem mining) ensures that version-space metrics are comparable across instances and reproducible across runs.
3. **Filters are essential.** Without atomic and subformula filters, a significant fraction of instances admit trivial solutions that inflate accuracy without testing quantifier reasoning.
4. **Generation success is not uniform.** Some gold formulas are harder to instantiate (valid worlds are rarer). We track generation failure rates and ensure that accepted instances are not biased toward easy-to-generate subfamilies.

## B.12. World Generation Hyperparameters

Table 28 lists the exact generation parameters used for each task and band in the v1 benchmark. These parameters are frozen in the released configuration files.

**Unary sampling**: Each element $a \in D$ is assigned $P(a) =$ true independently with probability $p_{\text{unary}} = 0.4$, subject to a balance constraint requiring 15–85% of the domain to satisfy each unary predicate.

**Binary sampling**: For regular out-degree mode, each element $a$ samples exactly 2 outgoing edges for $R$ (uniformly from $D \setminus \{a\}$) and exactly 2 outgoing edges for $S$, yielding expected edge density $\approx 2/|D|$ per relation.

**Unknown masking (EC)**: For each world, we collect all ground atoms of the unknown-eligible predicates, shuffle them, and mark the first $\lfloor$unknown_rate $\times$ total_atoms$\rfloor$ as unknown. The unknown rate is 20% for both EC bands; the core band masks atoms from $R$ and $S$, while the hard band masks only $R$ atoms.

*Table 28.* World generation hyperparameters (v1). $k$ = number of worlds per instance; unknown rate applies to EC only.

| Task | Band | Domain | $k$ | Unknown rate | Unknown preds |
|---|---|---|---|---|---|
| FullObs | simple | 5–7 | 4 | — | — |
| FullObs | easy | 5–7 | 6 | — | — |
| FullObs | medium | 7–10 | 8 | — | — |
| FullObs | hard | 8–12 | 10 | — | — |
| FullObs | extreme | 8–12 | 10 | — | — |
| CI | core | 7–9 | 7–8 YES + 2–3 NO | — | — |
| CI | lift_mix | 7–9 | 7–8 YES + 2–3 NO | — | — |
| EC | core | 6–8 | 3 | 20% | $R, S$ |
| EC | hard | 7–9 | 3 | 20% | $R$ only |

## B.13. EC Relevance Filter Pseudocode

The relevance filters ensure unknown atoms are genuinely relevant to the gold formula:

```
def check_relevance(world, formula, mode):
  C0 = complete_unknowns(world, False)
  C1 = complete_unknowns(world, True)
  m0 = matches_target(formula, world, C0)
  m1 = matches_target(formula, world, C1)
  if mode == "extreme_or":
    return m0 or m1
  elif mode == "middle_allowed":
    return not (m0 and m1)
```

`extreme_or` (core): at least one extreme completion works.

`middle_allowed` (hard): reject if both extremes work, which would make the instance trivial under extreme completions.

## B.14. Model and Decoding Settings

Table 29 lists the exact model identifiers and decoding parameters used for evaluation. All prompted models received the same prompt template (Appendix E). Each API call was retried up to 5 times with exponential backoff on transient failures. Remaining missing or unparsable outputs are reflected in the coverage columns of the main tables and count as incorrect under denominator-all reporting.

*Table 29.* Model identifiers and decoding settings. $T$ = sampling temperature; max_tok = maximum output tokens; think = reasoning/thinking token budget. A dash (—) indicates the provider default was used. Gemini 3.5f denotes Gemini 3.5 Flash; DSR denotes DeepSeek-Reasoner.

| Display Name | Provider / Model ID | $T$ | max_tok | think |
|---|---|---|---|---|
| GPT-5.4 | OpenAI / `gpt-5.4` | — | 64k | med |
| GPT-5.2 | OpenAI / `gpt-5.2` | — | 64k | med |
| Grok4 | XAI / `grok-4-0709` | 0.1 | 64k | — |
| Gemini 3.5f | Google / `gemini-3.5-flash` | — | — | med |
| Opus 4.6 | Anthropic / `claude-opus-4-6` | 0.1 | 64k | 32k |
| DeepSeek4Pro | DeepSeek / `deepseek-v4-pro` | 0.1 | 128k | high |
| Gemini 3.1 | Google / `gemini-3.1-pro-preview` | — | — | med |
| Gemini 3 | Google / `gemini-3-pro-preview` | — | — | med |
| Grok4.1f | XAI / `grok-4-1-fast-reasoning` | 0.1 | 64k | — |
| Grok4.3 | XAI / `grok-4.3` | 0.1 | 64k | med |
| DSR | DeepSeek / `deepseek-reasoner` | — | — | on |
| Opus 4.5 | Anthropic / `claude-opus-4-5-20251101` | 0.1 | 21k | — |
| Hermes4 | OpenRouter / `hermes4` | 0.1 | 32k | high |
| GPT-4o | OpenAI / `gpt-4o` | 0.1 | — | — |

All prompted models use JSON output format where supported. Temperature is set to 0.1 except for reasoning-native models (GPT-5.4, GPT-5.2, DeepSeek4Pro, DSR, and Gemini models) where the provider ignores or does not expose temperature. Extended thinking/reasoning is enabled for all models that support it; the "think" column reports the default token budget or effort level used.

### B.15. Equality Usage Analysis

The gold formulas in our benchmark do not use equality predicates—all target concepts are expressed using only the unary predicates $P, Q$ and binary predicates $R, S$. However, models are free to use equality $(= x\, y)$ in their predictions, and some models do so to express equivalent definitions or to construct case splits.

Tables 30 to 32 summarize equality usage across tasks. Key observations:

- Equality usage is strongly model-dependent rather than concentrated only in the strongest models. Opus 4.6 is the heaviest equality user in FullObs and EC (41.9% and 24.6%), while GPT-5.2 is highest in CI (28.0%).
- GPT-5.2 remains the most equality-heavy model among the top GPT models, especially in FullObs (26.7% usage, average AST 95.2), where equality often appears inside large case-split formulas.
- GPT-5.4 uses equality less often than GPT-5.2 but more successfully in EC: 9.5% of GPT-5.4 EC outputs contain equality and 84.2% of those formulas are valid, versus 12.5% usage and 76.0% validity for GPT-5.2.
- Aggregated over all models, equality-containing formulas are least reliable in FullObs (27.3% valid), more reliable in EC (46.6%), and most reliable in CI (78.0%), consistent with the contrastive setting admitting more alternative correct definitions.
- Several lower-accuracy models use equality sparingly; Hermes4 and GPT-4o almost never rely on it.

*Table 30.* **Equality usage in FullObs predictions.** *Total* = number of responses returned by model; *% Using = =* fraction of predictions containing equality; *Avg AST* = mean AST size of equality-containing formulas; *Valid %* = validity rate among equality-containing predictions.

| Model | Total | % Using = | Avg AST | Valid % |
|---|---|---|---|---|
| GPT-5.4 | 375 | 7.7% | 58.6 | 58.6% |
| GPT-5.2 | 375 | 26.7% | 95.2 | 55.0% |
| Grok4 | 335 | 5.1% | 52.4 | 52.9% |
| Gemini 3.5f | 374 | 1.9% | 22.7 | 0.0% |
| Opus 4.6 | 375 | **41.9%** | 31.9 | 12.7% |
| DeepSeek4Pro | 372 | 7.0% | 45.9 | 19.2% |
| Gemini 3.1 | 375 | 0.3% | 33.0 | **100.0%** |
| Gemini 3 | 375 | 9.9% | 32.0 | 13.5% |
| Grok4.1f | 369 | 5.4% | 19.2 | 15.0% |
| Grok4.3 | 342 | 1.8% | 16.8 | 16.7% |
| DSR | 374 | 2.9% | 24.3 | 18.2% |
| Opus 4.5 | 370 | 5.4% | 18.1 | 5.0% |
| Hermes4 | 373 | 0.8% | 9.0 | 0.0% |
| GPT-4o | 375 | 0.5% | 13.0 | 0.0% |
| *Total* | 5159 | 8.5% | 47.8 | 27.3% |

*Table 31.* **Equality usage in CI predictions.** *Total* = number of responses returned by model; *% Using = =* fraction of predictions containing equality; *Avg AST* = mean AST size of equality-containing formulas; *Valid %* = validity rate among equality-containing predictions.

| Model | Total | % Using = | Avg AST | Valid % |
|---|---|---|---|---|
| GPT-5.4 | 200 | 19.0% | 67.6 | 78.9% |
| GPT-5.2 | 200 | **28.0%** | 49.6 | 89.3% |
| Grok4 | 200 | 15.0% | 25.6 | 93.3% |
| Gemini 3.5f | 200 | 20.0% | 23.5 | 80.0% |
| Opus 4.6 | 200 | 26.0% | 27.6 | 69.2% |
| DeepSeek4Pro | 197 | 12.2% | 31.2 | 79.2% |
| Gemini 3.1 | 200 | 6.0% | 13.7 | **100.0%** |
| Gemini 3 | 200 | 12.5% | 21.9 | 76.0% |
| Grok4.1f | 195 | 17.4% | 19.9 | 85.3% |
| Grok4.3 | 200 | 5.5% | 20.1 | 63.6% |
| DSR | 198 | 6.6% | 23.7 | 69.2% |
| Opus 4.5 | 200 | 9.0% | 18.6 | 50.0% |
| Hermes4 | 199 | 2.0% | 18.0 | 0.0% |
| GPT-4o | 198 | 1.0% | 15.0 | 0.0% |
| *Total* | 2787 | 12.9% | 32.3 | 78.0% |

*Table 32*. **Equality usage in EC predictions.** *Total* = number of responses returned by model; *% Using* = = fraction of predictions containing equality; *Avg AST* = mean AST size of equality-containing formulas; *Valid %* = validity rate among equality-containing predictions.

| Model | Total | % Using = | Avg AST | Valid % |
|---|---|---|---|---|
| GPT-5.4 | 199 | 9.5% | 41.8 | **84.2%** |
| GPT-5.2 | 200 | 12.5% | 48.9 | 76.0% |
| Grok4 | 199 | 6.5% | 18.2 | 38.5% |
| Gemini 3.5f | 200 | 8.5% | 17.5 | 29.4% |
| Opus 4.6 | 199 | **24.6%** | 20.2 | 42.9% |
| DeepSeek4Pro | 199 | 7.5% | 26.6 | 40.0% |
| Gemini 3.1 | 200 | 2.0% | 13.8 | 75.0% |
| Gemini 3 | 200 | 7.0% | 23.7 | 35.7% |
| Grok4.1f | 190 | 6.3% | 16.6 | 58.3% |
| Grok4.3 | 200 | 4.0% | 15.2 | 50.0% |
| DSR | 200 | 4.0% | 16.0 | 37.5% |
| Opus 4.5 | 198 | 11.6% | 14.6 | 13.0% |
| Hermes4 | 200 | 0.5% | 13.0 | 0.0% |
| GPT-4o | 200 | 0.0% | — | — |
| *Total* | 2784 | 7.5% | 24.6 | 46.6% |

# C. Target Language and Formula Templates

This appendix specifies the bounded target language used for planted concepts. We first describe how structural templates are expanded into task-specific eligible target pools, and then list the core template families. The listed templates are representative structural patterns; the benchmark expands them through predicate choices, polarity choices, and argument-order variants. Each target formula has one free variable $x$; bound variables are drawn from $y, z$.

## C.1. Target-Pool Construction and Census

The benchmark uses a systematic bounded target language rather than exhaustive enumeration over raw first-order syntax. We first define a bounded target language by structural templates, then expand those templates into concrete formulas. Exhaustiveness is applied after this design choice: for each task or band, the generator enumerates all concrete formulas in the eligible pool induced by the task- or band-specific filters. The released benchmark is then a balanced subset of those eligible targets, paired with generated worlds that pass informativeness and shortcut-rejection filters.

The target-pool construction has the following stages.

1. Start from structural formula templates.
2. Instantiate predicate choices, polarity or guard choices where applicable, and argument-order variants.
3. Canonicalize and deduplicate the resulting concrete formulas.
4. Compute formula metadata: AST size, quantifier depth, family, subfamily, lift-hardness, and task-specific tags.
5. For each task or band, apply eligibility filters such as QD range, AST range, family/subfamily restrictions, lift-hard inclusion or exclusion, and EC family/relevance restrictions.
6. Enumerate the full resulting eligible pool.
7. Select planted gold formulas using balancing rules so that no single formula, family, or subfamily dominates.
8. Generate finite worlds for the selected gold formulas.
9. Reject generated instances solved by trivial shortcuts, including atomic, proper-subformula, or bounded quantifier-free shortcuts where those filters apply.

*Table 33.* Target-pool census. Counts are over concrete formulas obtained by expanding the structural template library through predicate and argument-order variants. "Eligible" means satisfying the task/band-specific structural and complexity filters before balanced gold selection.

| Task / band | Expanded before filters | Eligible after filters | Distinct planted golds | Released instances |
|---|---|---|---|---|
| FullObs simple | 160 | 148 | 25 | 25 |
| FullObs easy | 1254 | 60 | 49 | 100 |
| FullObs medium | 1254 | 100 | 46 | 100 |
| FullObs hard | 1254 | 186 | 82 | 100 |
| FullObs extreme_logic | 1254 | 186 | 25 | 25 |
| FullObs extreme_context | 1254 | 186 | 25 | 25 |
| CI core | 1254 | 116 | 65 | 120 |
| CI lift_mix | 1254 | 186 | 80 | 80 |
| EC core | 332 | 302 | 120 | 120 |
| EC hard | 1254 | 52 | 50 | 80 |

The rows differ because each task/band applies a different view of the expanded template library. FullObs simple uses the QD=1 pool; the remaining FullObs bands and both CI bands use the nested-template pool. EC core uses a separate expanded QD=1 pool, while EC hard uses the nested-template pool restricted to the EC-hard family set and non-lift targets. The approximately 200 figure in the main text refers to structurally distinct template-level formulas in the curated library. The expanded eligible pools can be larger because templates are instantiated with predicate choices and argument-order variants before task-specific filtering.

Gold selection is separate from eligibility. After each eligible pool is enumerated, the released benchmark selects planted formulas with balancing rules that control reuse across formulas, families, and subfamilies. In FullObs, band-level QD/AST filters are combined with family caps and lift-hard quotas. When a band requires more accepted instances than the number of distinct eligible planted formulas that yield accepted worlds, multiple independently generated instances may share the same planted formula. In CI, both reported bands draw from the QD=2, AST 12–18 pool: core selects non-lift formulas, while lift_mix includes a mixture of lift and non-lift targets. Both CI bands cap reuse at two instances per gold formula and four per subfamily.

EC uses task-specific eligible pools reflecting the two observation regimes: core draws from the expanded QD=1 pool, while

hard draws from the nested-template pool restricted to the EC-hard family set and non-lift targets. These choices were used to calibrate EC-core and EC-hard separately.

Target-pool enumeration is therefore separate from informative-world construction: the former defines and enumerates the candidate concepts, while the latter searches for finite structures that make the chosen concept nontrivial by eliminating plausible shortcuts.

The census above reports how many concrete targets are eligible for each task/band. The remainder of this appendix documents the structural templates underlying those pools.

### C.2. Formula Families

We classify templates into structural families according to quantifier pattern, guard structure, relation usage, and nesting. These tags are used both for balanced target selection and for structural error analysis. Table 34 summarizes the family tags.

| Family | Pattern Description |
|--------|---------------------|
| A | $\forall y(\neg\text{BIN}(x, y) \vee \exists z\, \text{BIN}(y, z))$ — simple $\forall\exists$ |
| B | $\exists y(U(y) \wedge \forall z \ldots)$ — guarded $\exists\forall$ with positive unary |
| C | $\exists y(\neg U(y) \wedge \forall z \ldots)$ — guarded $\exists\forall$ with negated unary |
| D | $\forall y(\neg\text{BIN}(x, y) \vee \exists z(\text{BIN} \wedge U))$ — $\forall\exists$ with unary filter |
| F | $\forall y(\neg\text{BIN}(x, y) \vee \exists z(\text{BIN} \wedge \neg U))$ — $\forall\exists$ with negated filter |
| G | $\exists y\forall z \ldots$ — simple $\exists\forall$ (no guard) |
| H | $\exists y\exists z \ldots$ — path/composition patterns (two existentials) |
| M | $\forall y(\neg\text{BIN}(x, y) \vee \exists z(\text{BIN}(y, z) \wedge \text{BIN}(x, z)))$ — mixed-witness |
| Z | $\exists y(U(y) \wedge \forall z(\neg\text{BIN}(y, z) \vee (\text{BIN}(x, z) \wedge V(z))))$ — $\exists\forall$ with $z$-filter |

*Table 34.* Formula family classification. BIN denotes a binary predicate ($R$ or $S$); $U, V$ denote unary predicates ($P$ or $Q$).

**Lift-hard patterns.** A formula is classified as lift-hard when a binary atom involving the free variable $x$ appears inside universal scope. For example, $\forall y(\neg R(x, y) \vee \exists z\, S(y, z))$ is lift-hard because $R(x, y)$ is evaluated under $\forall y$. These formulas require reasoning about all relevant neighbors of $x$, and they form a targeted stress test for universal relational generalization.

### C.3. Quantifier-Free Templates (QD=0)

QD=0 templates provide simple quantifier-free baselines used mainly for sanity checks and shortcut filtering.

```
(P x)
(Q x)
(not (P x))
(not (Q x))
(R x x)
(S x x)
(not (R x x))
(not (S x x))
(and (P x) (Q x))
(or (P x) (Q x))
(and (P x) (not (Q x)))
(or (not (P x)) (Q x))
(and (not (P x)) (not (Q x)))
(or (not (P x)) (not (Q x)))
```

### C.4. Single-Quantifier Templates (QD=1)

QD=1 templates introduce a single existential or universal quantifier, including edge-existence, guarded-existence, and guarded-universal patterns.

```
(exists y (R x y))
(exists y (S x y))
(exists y (R y x))
(exists y (S y x))
(exists y (and (R x y) (P y)))
(exists y (and (R x y) (Q y)))
(exists y (and (S x y) (P y)))
(exists y (and (R x y) (not (P y))))
(exists y (and (S x y) (not (Q y))))
(forall y (or (not (R x y)) (P y)))
```

```
(forall y (or (not (R y x)) (P y)))
(forall y (or (not (S x y)) (Q y)))
(forall y (or (not (R x y)) (Q y)))
(forall y (or (not (S y x)) (P y)))
(and (P x) (exists y (R x y)))
(and (not (P x)) (exists y (and (R x y) (Q y))))
(or (P x) (forall y (or (not (R x y)) (Q y))))
(and (Q x) (exists y (S x y)))
```

### C.5. Nested-Quantifier Templates (QD=2)

QD=2 templates introduce nested quantification, including $\exists\forall$, $\forall\exists$, and two-hop relational patterns. These templates are the main source of the harder structural families.

```
(exists y (forall z (or (not (R y z)) (S x z))))
(exists y (forall z (or (not (S y z)) (R x z))))
(exists y (and (P y) (forall z (or (not (R y z)) (S x z)))))
(exists y (and (Q y) (forall z (or (not (S y z)) (R x z)))))
(exists y (and (not (P y)) (forall z (or (not (R y z)) (S x z)))))
(forall y (or (not (R x y)) (exists z (S y z))))
(forall y (or (not (S x y)) (exists z (R y z))))
(forall y (or (not (R x y)) (exists z (and (S y z) (P z)))))
(forall y (or (not (R x y)) (exists z (and (S y z) (Q z)))))
(forall y (exists z (and (R x y) (S y z))))
(and (P x) (exists y (forall z (or (not (R y z)) (S x z)))))
(and (not (Q x)) (forall y (or (not (R x y)) (exists z (S y z)))))
(or (P x) (exists y (forall z (or (not (R y z)) (S x z)))))
(or (not (P x)) (forall y (or (not (R x y)) (exists z (S y z)))))
(and (Q x) (forall y (or (not (S x y)) (exists z (R y z)))))
(exists y (forall z (or (not (R z y)) (S z x))))
(forall y (or (not (R y x)) (exists z (S z y))))
```

The lists above show the core structural templates. The benchmark target pools expand these templates by swapping predicates, changing argument order, and instantiating guard/polarity choices, followed by deduplication and task-specific filtering as described in Appendix C.1.

# D. Task Semantics and Qualitative Examples

In this appendix, concrete formulas illustrate the benchmark semantics and the main qualitative behaviors of prompted models. A toy target concept first shows how FullObs, CI, and EC interpret the same first-order output language. Benchmark examples then show compact formulas, bloated case-splitting formulas, and structurally plausible formulas that fail by dropping or weakening a required relational pattern.

## D.1. Task Semantics Examples

The same first-order formula is subject to different semantic requirements in the three tasks. We illustrate the distinction with the toy target

$$\varphi^\star(x) = P(x) \land \exists y\, R(x, y).$$

**FullObs.** In FullObs, all non-target facts in the input worlds are observed. In a world with domain $\{a, b\}$, suppose $P(a)$ is true, $P(b)$ is false, $R(a, b)$ is true, and all other relevant ground atoms are false. Then $\varphi^\star$ selects $a$ and rejects $b$. A FullObs solution must reproduce the target extension in every input world.

**CI.** CI uses the same exact-match condition on YES worlds, but adds contrastive NO worlds. Let $\delta(x) = P(x)$ be a shortcut. YES worlds can be chosen so that every $P$-object also has an $R$-successor, making $\delta$ and $\varphi^\star$ agree there. A NO world can instead be labeled by $\delta$ while containing a $P$-object with no $R$-successor. Then $\delta$ exactly matches the NO target and is rejected by the CI criterion, whereas $\varphi^\star$ does not exactly match that NO target. Thus a CI NO world is not a label inversion; it is a target extension that the candidate formula must avoid matching exactly.

**EC.** EC hides some non-target facts. Let the domain be $\{a, b\}$, let $P(a)$ be known true, $P(b)$ known false, $R(a, b)$ unknown, and all other $R$-facts known false. If the target labels $a$ positive and $b$ negative, then $\varphi(x) = P(x) \land \exists y\, R(x, y)$ is compatible with the observations: a completion setting $R(a, b)$ to true makes $a$ satisfy the formula, while $b$ cannot satisfy it because $P(b)$ is known false. EC asks for one completion per world, not one global completion shared across worlds and not validity under all completions.

## D.2. FullObs Examples

FullObs gives the cleanest setting for the bloat/generalization contrast: a formula can satisfy every input world either by recovering a compact relational rule or by fitting the finite instance with a much larger case analysis.

**Compact successful formulas.** FullObs simple 006 illustrates that an instance-correct formula can be shorter than the planted reference. Several prompted models return:

| Model | AST | Formula | $\Delta$ |
|---|---|---|---|
| *Gold* | 20 | `(and (and (not (P x)) (not (Q x))) (exists y (and (R x y) (and (P y) (not (Q y))))))` | — |
| Grok4.1f | 12 | `(and (not (or (P x) (Q x))) (exists y (S x y)))` | $-8$ |
| Grok4 | 13 | `(and (not (P x)) (not (Q x)) (exists y (S x y)))` | $-7$ |
| GPT-5.2 | 13 | `(and (not (P x)) (not (Q x)) (exists y (S x y)))` | $-7$ |

The returned formulas show that, on these finite input worlds, a simpler $S$-based condition can replace the planted $R$-based witness condition. This illustrates why planted gold formulas are reference concepts rather than guaranteed shortest equivalents on sampled worlds.

**Paired parsimony contrast.** FullObs easy 005 gives a paired contrast between compact recovery and over-budget fit. The planted formula has AST 15:

```
(forall y (or (not (S x y)) (exists z (and (S y z) (Q z)))))
```

GPT-5.4 returns the same formula, with AST 15 and $\Delta = 0$; it is instance-correct and budget-compliant. GPT-5.2 is also instance-correct on the same input worlds, but returns a 70-node formula with $\Delta = +55$, exceeding the gold+25 budget:

```
(and (not (S x x)) (forall y (or (not (S x y)) (exists z (S y z)))) (or (exists y (S y x)) (forall y (or
(not (R x y)) (exists z (S y z))))) (or (not (exists y (S x y))) (exists y (exists z (and (R x y) (R x z)
(not (= y z)))))) (and (Q x) (not (exists y (S y x))))))
```

| Model | Status | AST | $\Delta$ |
|---|---|---|---|
| GPT-5.2 | instance-correct, over-budget | 70 | $+55$ |
| GPT-5.4 | instance-correct, budget-compliant | 15 | 0 |

This pair illustrates why budgeted metrics are informative: both formulas satisfy the input worlds, but only the compact

formula recovers the planted universal-existential rule.

**Valid but bloated formula.** FullObs hard 082 shows extreme case-splitting. The planted formula is a compact universal condition over all $S$-successors of $x$ (gold AST 15):

*Gold formula:*

```
(forall y (or (not (S x y)) (exists z (and (R y z) (Q z)))))
```

GPT-5.2 returns an instance-correct 370-node formula ($\Delta = +355$) that branches on global properties of the input worlds:

*GPT-5.2 prediction (AST=370, $\Delta = +355$, instance-correct but over-budget):*

```
(or (and (not (exists y (not (exists z (R y z)))))) (not (exists y (not (exists z (S y z)))))) (or (and (P
x) (Q x)) (and (not (Q x)) (exists y (and (R x y) (Q y))) (or (forall z (or (not (R x z)) (P z))) (exists
w (and (R x w) (Q w) (not (P w))))))))))) (and (exists y (not (exists z (R y z)))) (not (exists y (not
(exists z (S y z))))) (and (Q x) (forall y (or (not (S x y)) (Q y))))) (and (exists y (not (exists z (R y
z)))) (exists y (not (exists z (S y z)))) (exists y (and (Q y) (not (P y)) (not (exists z (S y z))))) (or
(not (Q x)) (P x) (not (exists y (S x y)))))) (and (exists y (not (exists z (R y z)))) (exists y (not
(exists z (S y z)))) (not (exists y (and (Q y) (not (P y)) (not (exists z (S y z)))))) (and (not (Q x))
(exists y (exists z (exists w (and (R x y) (R x z) (R x w) (not (= y z)) (not (= y w)) (not (= z
w))))))))) (and (not (exists y (not (exists z (R y z))))) (exists y (not (exists z (S y z)))) (not (exists
y (S y y))) (not (exists y (S x y)))) (and (not (exists y (not (exists z (R y z))))) (exists y (not
(exists z (S y z)))) (exists y (S y y)) (exists y (and (Q y) (not (P y)))) (or (not (P x)) (and (Q x)
(forall y (or (not (R x y)) (P y)))) (exists y (and (R x y) (Q y) (not (P y)))) (and (P x) (not (S x x))
(exists y (and (Q y) (S y x) (not (= y x))))))))) (and (not (exists y (not (exists z (R y z))))) (exists y
(not (exists z (S y z)))) (exists y (S y y)) (not (exists y (and (Q y) (not (P y)))))) (Q x)))
```

The formula is correct on the input instance but far outside the gold-relative budget. Its size and branching structure make the failure mode visible: exact input fit is achieved by finite-world case analysis rather than recovery of the compact rule.

**Invalid structural simplifications.** The invalid FullObs examples below show plausible but incomplete simplifications of universal structure.

| Instance | Model | Prediction | Gold formula | Failure |
|---|---|---|---|---|
| FullObs easy 003 | GPT-5.2 | `(forall y (or (not (R x y))`
`(Q y)))` | `(forall y (or (not (R x y))`
`(exists z (and (S y z) (P z)))))` | fails 3/4 input worlds |
| FullObs easy 005 | Grok4 | `(not (exists y (and (S x y)`
`(Q y))))` | `(forall y (or (not (S x y))`
`(exists z (and (S y z) (Q z)))))` | fails 2/4 input worlds |

In easy 003, GPT-5.2 replaces the required second-hop witness $\exists z \, (S(y, z) \land P(z))$ with the unary condition $Q(y)$, which matches one input world but fails on the others. In easy 005, Grok4 collapses the gold universal-existential requirement into a local negated existential, missing the condition that every relevant $S$-successor must have a further $S$-successor satisfying $Q$.

### D.3. CI Examples

CI illustrates a different tension: contrastive correctness can admit compact alternatives to the planted concept, but it does not by itself prevent large finite-world case splits.

**Compact successful formulas.** CI can admit compact alternatives because the YES/NO criterion is discriminative: a formula may satisfy the contrastive task without recovering the planted gold concept exactly. On CI core 047, GPT-5.2 and Grok4 return:

| Model | AST | Formula | $\Delta$ |
|---|---|---|---|
| *Gold* | 18 | `(exists y (and (or (P y) (Q y)) (forall z (or (not`
`(S y z)) (S z x)))))` | — |
| GPT-5.2 | 8 | `(or (exists y (R x y)) (Q x))` | $-10$ |
| Grok4 | 8 | `(or (P x) (exists y (R x y)))` | $-10$ |

These compact alternatives satisfy the same contrastive instance even though they differ from the planted formula.

CI core 026 gives another compact success. The planted formula has AST 17:

```
(exists y (exists z (and (and (R x y) (S y z)) (and (P z) (Q z)))))
```

Gemini 3.5 Flash returns:

```
(exists y (and (R x y) (exists z (and (S y z) (P z)))))
```

The prediction has AST 14 and $\Delta = -3$. It is CI-valid: it matches all YES worlds and avoids exact matches on the NO targets. The finite CI constraints do not require the extra $Q$ literal from the planted formula.

**Valid but bloated formula.** CI core 095 shows that contrastive correctness alone does not prevent case-splitting. The planted formula is a concise existential-universal rule (gold AST 16):

*Gold formula:*

```
(exists y (and (not (P y)) (forall z (or (not (R y z)) (R z x)))))
```

GPT-5.2 returns an instance-correct 539-node formula ($\Delta = +523$) that satisfies the CI input constraints through a large disjunction over special finite-world configurations:

*GPT-5.2 prediction (AST=539, $\Delta = +523$, instance-correct but over-budget):*

```
(or (and (exists y (and (R y y) (forall z (not (S y z))))) (exists y (and (R y y) (forall z (not (S y
z))) (R y x)))) (and (not (exists y (and (R y y) (forall z (not (S y z)))))) (exists y (and (Q y) (forall
z (or (not (Q z)) (= z y))))) (or (P x) (Q x) (forall y (not (S x y)))) (and (not (exists y (and (R y y)
(forall z (not (S y z)))))) (not (exists y (and (Q y) (forall z (or (not (Q z)) (= z y)))))) (exists y
(and (R y y) (not (P y)) (not (Q y)))) (exists y (and (R y y) (not (P y)) (not (Q y)) (= x y)))) (and
(not (exists y (and (R y y) (forall z (not (S y z)))))) (not (exists y (and (Q y) (forall z (or (not (Q
z)) (= z y)))))) (not (exists y (and (R y y) (not (P y)) (not (Q y))))) (exists y (and (Q y) (not (P y))
(R y y) (forall z (or (not (and (Q z) (not (P z)) (R z z))) (= z y)))) (or (and (= x y) (exists w (and (S
y w) (P w) (Q w)))) (and (R y x) (not (Q x)))))))) (and (not (exists y (and (R y y) (forall z (not (S y
z)))))) (not (exists y (and (Q y) (forall z (or (not (Q z)) (= z y)))))) (not (exists y (and (R y y) (not
(P y)) (not (Q y)))) (exists y (and (Q y) (not (P y)) (R y y) (forall z (or (not (and (Q z) (not (P
z)) (R z z))) (= z y)))))) (forall y (or (not (Q y)) (P y))) (and (P x) (not (exists w (and (P w) (not (Q
w)) (R w w) (forall z (or (not (and (P z) (not (Q z)) (R z z))) (= z w))) (Q x) (S x w) (forall y (or
(not (and (S x y) (not (Q y))) (= y w))))))))) (and (not (exists y (and (R y y) (forall z (not (S y
z)))))) (not (exists y (and (Q y) (forall z (or (not (Q z)) (= z y)))))) (not (exists y (and (R y y) (not
(P y)) (not (Q y)))) (not (exists y (and (Q y) (not (P y)) (R y y) (forall z (or (not (and (Q z) (not (P
z)) (R z z))) (= z y)))))) (not (forall y (or (not (Q y)) (P y)))) (exists y (and (Q y) (forall z (or
(not (R y z)) (not (Q z)))) (forall w (or (not (and (Q w) (forall z (or (not (R w z)) (not (Q z)))))) (=
w y))) (or (= x y) (R y x))))))
```

Although CI bloat is smaller on average than FullObs bloat, the contrastive YES/NO criterion alone does not rule out extreme case-splitting.

**Invalid structural simplifications.** The invalid CI examples show failures to satisfy the YES worlds. In these cases, the returned formulas use plausible local relations but miss the relational pattern required by the planted concept.

| Instance | Model | Prediction | Gold formula | Failure |
|---|---|---|---|---|
| CI core 002 | Gemini 3 | `(exists y (and (R x y) (not (= x y)) (or (not (S x y)) (and (not (exists z (R y z))) (not (P y))))))` | `(exists y (exists z (and (and (R x y) (S y z)) (not (P z)))))` | YES fail |
| CI core 005 | Gemini 3 | `(exists y (and (S x y) (not (Q y))))` | `(forall y (or (not (R x y)) (exists z (and (S y z) (P z)))))` | YES fail |

In CI core 002, Gemini 3 uses a local condition involving $R(x, y)$, disequality, and missing $S$-successors, but the planted concept requires a two-hop $R/S$ witness ending in $\neg P$. In CI core 005, Gemini 3 uses $S$ and $Q$ instead of the required $R$, $S$, and $P$ structure, missing the universal-existential pattern.

### D.4. EC Examples

EC changes the success criterion: a weaker formula may be completion-compatible, a bloated formula may become feasible, and a structural shortcut may still fail.

**Compact successful formulas.** EC can make weaker formulas sufficient because unknown atoms may be completed in ways that support the target labels. EC core 0010 admits a compact completion-compatible formula:

| Model | AST | Formula | $\Delta$ |
|---|---|---|---|
| *Gold* | 16 | `(and (not (P x)) (exists y (and (R x y) (and (P y) (not (Q y))))))` | — |
| Grok4.1f | 9 | `(and (not (P x)) (exists y (R x y)))` | −7 |
| DSR | 9 | `(and (not (P x)) (exists y (R x y)))` | −7 |
| GPT-5.2 | 9 | `(and (not (P x)) (exists y (R x y)))` | −7 |
| Gemini 3 | 9 | `(and (not (P x)) (exists y (R x y)))` | −7 |
| Opus 4.5 | 9 | `(and (not (P x)) (exists y (R x y)))` | −7 |

Under EC semantics, the detailed witness constraint $P(y) \wedge \neg Q(y)$ is not needed on these input worlds; the weaker condition of having an $R$-successor is compatible with the observed target labels under some completion.

**Universal-to-existential collapse.** EC hard 0071 illustrates the universal-to-existential collapse identified in Appendix B.5. The planted formula requires every relevant neighbor to have a second-hop witness, but the returned formula asks only for one local witness.

The planted formula has AST 18:

```
(forall y (or (not (R x y)) (exists z (and (R y z) (and (P z) (Q z))))))
```

DeepSeek4Pro returns (AST 8, $\Delta = -10$; invalid under EC semantics, failing all 3 input worlds):

```
(exists y (and (R x y) (P y)))
```

The error is structural: the returned formula replaces a condition over every $R$-successor of $x$ with the existence of one local $R$-neighbor satisfying $P$, dropping both the universal requirement and the second-hop witness.

**Valid but bloated formula.** EC core 0003 shows a smaller instance of the bloat phenomenon. The gold formula expresses a symmetric constraint on the $S$ relation:

*Gold formula:*

```
(forall y (or (or (not (S x y)) (not (S y x))) (and (P y) (Q y))))
```

GPT-5.2 returns a valid but much larger formula that enumerates finite-world configurations:

*GPT-5.2 prediction (AST=92, $\Delta = +75$, valid but over-budget):*

```
(or (and (exists y (and (P y) (Q y) (S y y))) (and (Q x) (or (not (S x x)) (P x)))) (and (not (exists y
(and (P y) (Q y) (S y y)))) (exists z (and (P z) (R z z))) (not (exists y (R y x)))) (and (not (exists y
(and (P y) (Q y) (S y y)))) (not (exists z (and (P z) (R z z)))) (and (Q x) (not (S x x)) (not (exists y
(and (R x y) (Q y)))))))
```

Under existential completion, feasibility can be easier than finding a succinct explanation; this example shows how a completion-compatible formula can still be far from the planted reference concept. GPT-5.2's prediction is valid on the EC input instance, but it has AST 92 ($\Delta = +75$) and is over the gold+25 budget.

**Invalid structural simplifications.** The invalid EC examples show that completion flexibility does not repair every plausible shortcut. Some returned formulas ignore the relation used by the gold concept or add disjuncts that cannot be made compatible with all input worlds.

| Instance | Model | Prediction | Gold formula | Failure |
|---|---|---|---|---|
| EC core 0001 | Grok4 | `(and (P x) (or (exists y (and (R x y) (Q y))) (exists z (and (R x z) (S z z)))))` | `(and (P x) (exists y (and (R x y) (Q y))))` | invalid |
| EC core 0003 | Grok4 | `(not (R x x))` | `(forall y (or (or (not (S x y)) (not (S y x))) (and (P y) (Q y))))` | invalid |

In EC core 0001, Grok4 adds an unnecessary disjunct involving $S(z, z)$, which cannot be made compatible with all input worlds under existential completion. In EC core 0003, Grok4 returns $\neg R(x, x)$, ignoring the $S$-relation structure that defines the target.

# E. Prompts

This section presents the prompts used for each of the three induction tasks. Each prompt consists of a task description followed by the problem instance (worlds and target labels), and concludes with output format instructions. The prompts below show the task-specific portions; the problem instance is inserted between the task description and the output instructions. For FullObs, one example problem description is shown.

## E.1. FullObs Prompt

```
# First-Order Logic Concept Synthesis

## Task Overview

You are given several finite "worlds," each containing:
- A finite domain of objects (named a0, a1, a2, ...)
- Interpretations of predicates (which objects/pairs satisfy each predicate)
- A target concept T(x) that specifies which objects are "positive" (T is TRUE)
  and which are "negative" (T is FALSE)

**Closed World Assumption**: Only the facts explicitly listed as TRUE are true.
Any predicate application (P(a), R(a,b), etc.) not explicitly listed should be
assumed FALSE.

**Your goal**: Find a first-order logic formula phi(x) with one free variable x
that **perfectly separates** the positive and negative examples:
- phi(c) must evaluate to TRUE for every object c where T(c) is TRUE
- phi(c) must evaluate to FALSE for every object c where T(c) is FALSE

The formula must work correctly for ALL objects in ALL input worlds.

## Output Format

You must output your formula in S-expression syntax. The grammar is:

phi ::= (P x)              -- unary predicate applied to variable
      | (R x y)            -- binary predicate applied to two variables
      | (= x y)            -- equality of two variables
      | (not phi)          -- negation
      | (and phi1 phi2)    -- conjunction (2 or more arguments)
      | (or phi1 phi2)     -- disjunction (2 or more arguments)
      | (forall v phi)     -- universal quantification
      | (exists v phi)     -- existential quantification

**Important constraints:**
- Your formula must have exactly one free variable: x
- All other variables must be bound by quantifiers (forall or exists)
- Variable names should be: x (free), y, z, w (bound by quantifiers)
- Prefer simpler formulas; avoid redundant conjuncts and case-splitting

## Problem Instance

## Training Worlds (learn from these):

### World: train_0
Domain: {a0, a1, a2, a3, a4, a5, a6}

**Predicates:**
- P: a0, a1, a2, a3, a4
- Q: a0, a1, a3, a6
- R: (a0, a0), (a0, a4), (a1, a5), (a1, a6), (a2, a5), (a2, a6), (a3, a1), (a3, a2),
    (a4, a2), (a5, a0), (a5, a2), (a5, a3), (a5, a4), (a6, a1), (a6, a3), (a6, a5)
- S: (a0, a1), (a0, a6), (a1, a0), (a1, a5), (a1, a6), (a2, a1), (a3, a0), (a3, a2),
    (a3, a3), (a3, a6), (a4, a1), (a4, a4), (a5, a5), (a5, a6), (a6, a2), (a6, a6)
```

```
**Target T(x):**
- T is TRUE for: {a0, a5, a6}
- T is FALSE for: {a1, a2, a3, a4}
```

### World: train_1
Domain: {a0, a1, a2, a3, a4, a5, a6, a7}

```
**Predicates:**
- P: a0, a2, a3, a5, a7
- Q: a1, a3, a4, a6, a7
- R: (a0, a2), (a0, a3), (a1, a1), (a1, a2), (a1, a5), (a1, a6), (a2, a0), (a2, a3),
     (a2, a6), (a2, a7), (a3, a0), (a4, a2), (a5, a0), (a5, a2), (a5, a3), (a5, a5),
     (a5, a7), (a6, a0), (a6, a1), (a6, a3), (a7, a4)
- S: (a0, a1), (a1, a1), (a1, a2), (a1, a4), (a1, a5), (a2, a2), (a2, a7), (a3, a3),
     (a3, a6), (a4, a1), (a4, a4), (a5, a2), (a5, a4), (a6, a0), (a6, a2), (a7, a0),
     (a7, a2), (a7, a5)
```

```
**Target T(x):**
- T is TRUE for: {a1, a2, a4, a5}
- T is FALSE for: {a0, a3, a6, a7}
```

### World: train_2
Domain: {a0, a1, a2, a3, a4, a5, a6}

```
**Predicates:**
- P: a0, a2, a4, a6
- Q: a3, a4, a6
- R: (a0, a0), (a0, a1), (a0, a2), (a0, a6), (a1, a1), (a1, a2), (a2, a4), (a3, a0),
     (a3, a1), (a3, a3), (a3, a5), (a3, a6), (a5, a0), (a5, a1), (a5, a3), (a6, a4),
     (a6, a6)
- S: (a0, a0), (a1, a0), (a1, a2), (a1, a4), (a2, a2), (a2, a5), (a3, a1), (a4, a0),
     (a4, a1), (a4, a6), (a5, a2), (a5, a3), (a5, a4), (a6, a0), (a6, a1), (a6, a4),
     (a6, a6)
```

```
**Target T(x):**
- T is TRUE for: {a0, a2}
- T is FALSE for: {a1, a3, a4, a5, a6}
```

### World: train_3
Domain: {a0, a1, a2, a3, a4, a5, a6, a7}

```
**Predicates:**
- P: a0, a1, a2, a3, a4
- Q: a0, a3, a7
- R: (a0, a5), (a1, a1), (a1, a4), (a1, a5), (a2, a2), (a2, a3), (a3, a2), (a3, a6),
     (a4, a3), (a5, a0), (a5, a1), (a5, a5), (a5, a7), (a6, a7), (a7, a3), (a7, a7)
- S: (a0, a2), (a0, a7), (a1, a2), (a1, a4), (a2, a1), (a2, a2), (a2, a6), (a3, a1),
     (a3, a4), (a4, a0), (a4, a2), (a4, a5), (a4, a6), (a5, a0), (a5, a6), (a5, a7),
     (a6, a0), (a6, a1), (a6, a2), (a7, a1), (a7, a3)
```

```
**Target T(x):**
- T is TRUE for: {a1, a2, a6}
- T is FALSE for: {a0, a3, a4, a5, a7}
```

## Your Task

Analyze the input worlds carefully. Identify what **distinguishes** objects
where T is TRUE from objects where T is FALSE.

**Think step by step:**
1. For each input world, compare the T-TRUE objects against the T-FALSE objects
2. Look for properties (unary predicates P, Q) or relationships (binary predicates
   R, S) that correlate with T

3. Check: Do all T-TRUE objects share some property? Do all T-FALSE objects lack it?
4. Consider whether the pattern involves existential quantification
   ("there exists a y such that...") or universal quantification ("for all y...")
5. Formulate your hypothesis as an S-expression formula
6. Verify: For each object in each input world, check that your formula gives
   TRUE exactly when T is TRUE

## Output

Your final answer must be exactly ONE LINE containing ONLY valid JSON:
- "formula": a single S-expression formula string with one free variable x
- "description": a short plain-English description (one sentence)

Example:
{"formula":"(exists y (and (R x y) (P y)))","description":"x has an R-successor
that satisfies P."}

## E.2. CI (Contrastive) Prompt

# First-Order Logic Concept Synthesis (Zendo-Style)

## Task Overview

You are given two sets of finite "worlds":
- **YES worlds**: Worlds where the hidden rule is satisfied
- **NO worlds**: Worlds where the hidden rule is NOT satisfied

Each world contains:
- A finite domain of objects (named a0, a1, a2, ...)
- Interpretations of predicates (which objects/pairs satisfy each predicate)
- A target concept T(x) that labels certain objects as "positive examples"

**Closed World Assumption**: Only the facts explicitly listed as TRUE are true.

Your goal is to find a first-order logic formula phi(x) with one free variable x
such that:
- In YES worlds: phi(x) **exactly matches** T(x) for all objects
- In NO worlds: phi(x) **fails to match** T(x) -- at least one object is misclassified

Think of this like the game Zendo: you must find the secret rule that all YES
worlds follow but NO worlds violate.

## Validity Criterion (Important)

Define **Match(W, phi)** := for all a in domain(W): W |= phi(a) iff a in T_true(W)

Your formula is **correct** iff:
1. For every YES world W: Match(W, phi) is TRUE
2. For every NO world W: Match(W, phi) is FALSE

This means your formula must work perfectly in YES worlds, and must have at least
one error in each NO world.

## Output Format

<same as FullObs prompt>

**Important constraints:**

<same as FullObs prompt>

[... Problem instance inserted here ...]

## Your Task

```
Analyze the YES and NO worlds carefully. Find the pattern that:
1. Perfectly matches T in all YES worlds
2. Fails to match T in all NO worlds
```

```
## Output
```

```
<same as FullObs prompt>
```

## E.3. EC (Existential Completion) Prompt

*(Corrected prompt used for all EC results reported.)*

```
# First-Order Logic Concept Synthesis (Partial Observation)
```

```
## Task Overview
```

```
You are given several finite "worlds" with **partial observations**:
- Some predicate facts are **known** (observed as TRUE or FALSE)
- Some predicate facts are **unknown** (not observed)
```

```
Each world contains:
- A finite domain of objects (named a0, a1, a2, ...)
- Known facts: predicates whose truth values have been observed
- Unknown atoms: predicates whose truth values are hidden
- A target concept T(x) that specifies which objects are "positive" (T is TRUE)
  and which are "negative" (T is FALSE)
```

```
**Observation Rules**:
- Atoms listed under "Known Facts" with explicit TRUE values are TRUE
- Atoms listed under "Unknown Atoms" have unknown truth values
- Any atom NOT listed as TRUE and NOT listed as Unknown is FALSE
```

```
The target T(x) values are always fully specified (not unknown).
```

```
**Your goal**: Find a first-order logic formula phi(x) with one free variable x
that **perfectly separates** the positive and negative examples:
- phi(c) must be TRUE for every object c where T(c) is TRUE
- phi(c) must be FALSE for every object c where T(c) is FALSE
```

```
**Completion semantics**: For each world separately, there must exist at least one
assignment of truth values to the unknown atoms such that phi matches T for all
objects in that world.
```

```
## Output Format
```

```
<same as FullObs prompt>
```

```
**Important constraints:**
```

```
<same as FullObs prompt>
```

```
[... Problem instance inserted here ...]
```

```
## Your Task
Analyze the input worlds carefully, keeping in mind that some facts are unknown.
```

```
**Think step by step:**
1. For each input world, compare objects where T is TRUE vs. T is FALSE
2. Note which predicate facts are known vs unknown
3. Look for patterns that **distinguish** T-TRUE objects from T-FALSE objects
   using the known facts
4. Consider whether the pattern involves existential or universal quantification
5. Formulate your hypothesis as an S-expression formula
6. Verify: Check that your formula gives TRUE exactly when T is TRUE, and FALSE
   exactly when T is FALSE (under some valid completion of unknowns)
```

**Key insight**: Your formula should work based on the known facts. Focus on patterns that depend on observed predicates.

## Output

<same as FullObs prompt>

