# OpenReview forum: "INDUCTION: Finite-Structure Concept Synthesis in First-Order Logic"
_ICML.cc/2026/Conference — ICML 2026 spotlight_

### Official Review · Reviewer_xyir · 2026-02-26

**Soundness:** 3
**Presentation:** 2
**Significance:** 4
**Originality:** 4
**Overall Recommendation:** 5
**Confidence:** 3

**Summary:**

The authors introduce the INDUCTION benchmarking suite. The basic task is to extract a first-order formula in one free variable that succinctly captures an abstract monadic predicate given a logical vocabulary and a set of examples (sometimes both positive and negative examples) over small finite structures. Three regimes are provided: (i) Positive examples only (FullObs), (ii) Positive and negative examples (CI - for Contrastive), (iii) Partial observations (EC - for the method of evaluation using Existential Completion semantics) - where some ground terms are unknown. The authors describe their difficulty-graded generation process for each of the three regimes. They offer evaluation metrics, combining validity with an assessment of formula complexity. In addition, they provide an extensive benchmarking of many contemporary large language and reasoning models on their baseline task(s) and provide all source code and datasets needed to reproduce the results of the paper.

**Compliance With Llm Reviewing Policy:**

Affirmed.

**Key Questions For Authors:**

1. I did not understand the EC model completely. When you say that "some ground atoms are unknown" do you mean that some values of the predicates are not revealed? Also, I do not know precisely what "existential completion semantics" is -- can you give examples?

2. You say that your "generators maintain pools of plausible distractor hypotheses and construct worlds to eliminate them." You go on to say that you "develop a targeted 'trap' mechanism that makes NO worlds informative". Can you give examples of these distractor hypotheses and worlds to eliminate them? Can you also provide more details about this targeted trap mechanism? I found the words here and elsewhere to be suggestive but hard to pin down.

3. How exactly were the gold formulas generated and how do you know that they are optimal? Perhaps it is sufficient that they are "close enough to optimal". Please comment and if close enough to optimal is the argument, please indicate how you know that the gold formulas are even close enough to optimal.

4. There were many other expressions that I felt needed to be precisely defined, but were not. In addition to answering my specific questions about particular formal definitions, I would like to know how the authors would modify their submission to make the details of the paper more apparent (e.g., use the formal latex definition environment). In particular, I think a lot of the detailed benchmarking results can be moved completely to the appendix. The presentation and detailed discussion of these results do not add that much to the presentation. It is enough to know that a detailed benchmarking was performed. Some particular issues: (i) What is meant precisely by the word "uniformly" at the end of the first paragraph of section 2? Does this mean that it defines T across all potential finite worlds? All worlds of a given size? It does not seem to have the usual complexity theoretic meaning (e.g., in descriptive complexity when one talks about a "uniform family of first order formulas" - here you are just talking about a single formula being uniform). (ii) What is meant by "high-capacity models" at the end of section 2? (iii) At the beginning of section 4.1, you talk about subfamilies and give some examples of attributes of the subfamilies, but I did not understand what is meant by "guard predicates"  - perhaps illustrate with some examples of these subfamilies. (iv) Later in section 4.1 you say that you use "lift-hard formulas" to create "headroom slices" in each task, but I have no idea what the phrases I have put in quotes here mean.

5. One of your syntactic measures is quantifier depth. In the subsequent discussion in section 4.1, it seemed like you were measuring this in terms of quantifier alternations (\forall \exists alternations). Is that the case, or is you quantifier depth purely what is called the quantifier rank in the literature? If it is the number of alternations, do you take the maximum number of alternations across any path from outer-most to inner-most quantifier, or do something else?

**Limitations:**

Yes

**Strengths And Weaknesses:**

I found the paper to be strong in most respects. It is both significant and highly original. It seems to fill an important niche - the production of a first-order formula to summarize a set of examples, with the examples coming from manageable-sized finite structures. I have not seen  anything in the literature that is very close to what is presented in this paper. The writing in the paper is good in most respects; it is very literate and well structured. I believe the paper is sound. Although there are no real theoretical results presented, the manner in which benchmark cases are generated was sometimes left a bit opaque (e.g., the generation of the gold target formulas and proof that these are optimal, which is never discussed), so it was difficult to assess if the benchmarks might be somehow biased. I found the main weakness of the paper to be that there are numerous undefined terms and no examples of formulas or small worlds presented in the main text, thus, for the reviewer not having time to study the appendices, it is only possible to read the paper at a gist or surface level. The paper itself does not even provide pointers to an Appendix where examples are given.

---

> ### Author Rebuttal · Authors · 2026-03-30
>
> Thank you for the thoughtful and encouraging review. We appreciate your view that the benchmark is highly original, fills an important niche and that the paper is strong overall. We also agree with your main criticism: the presentation should have made the technical details and examples much easier to access. In the revised manuscript, we will surface a small worked example in the main text, add explicit pointers to the appendix, move more low-level benchmarking detail out of the core narrative, and define overloaded terms more formally, including using more explicit formal definitions where appropriate.
>
> 1. EC: Yes, 'some ground atoms are unknown' means that some predicate values are hidden rather than revealed. A candidate formula is valid if, for each world independently, there exists a completion of those hidden atoms that makes the formula match the target extension in that world. Thus EC is world-local existential completion, not freedom to change facts globally. For example, in a one-object world $\{a0\}$ where $P(a0)$ and $Q(a0)$ are unknown and the target labels $a0$ positive, $P(x)\land \neg Q(x)$ is valid because the completion $P(a0)=\mathrm{T}, Q(a0)=\mathrm{F}$ makes it true on $a0$.
>
> 2. Distractors and traps. We agree this needed a more concrete explanation. In the generator, CI builds a pool of plausible shortcut hypotheses rather than arbitrary wrong formulas: unary atoms/negations such as $P(x), \neg P(x), Q(x)$; edge-existence tests such as $\exists yR(x,y)$ and $\exists yR(y,x)$; “no-edge” formulas such as $\forall y \neg R(x,y)$; guarded universals such as $\forall y (R(x,y)\to P(y))$; self-loops such as $R(x,x)$ and $\neg R(x,x)$; and, in harder variants, near-miss mutants of the gold formula such as dropped guards, argument swaps, and predicate swaps. YES worlds are accepted only if enough distractors still survive them; NO worlds are then built specifically to eliminate those surviving shortcuts.
>
> A minimal example is
> $\phi^\star(x)=P(x)\land \exists y R(x,y)$
> with distractors
> $\delta_1(x)=P(x)$ and $\delta_2(x)=\exists y R(x,y)$.
> If the YES worlds are chosen so that every $P$-object also has an $R$-successor, all three formulas survive YES. A NO world can then kill $\delta_1$ by containing an object that satisfies $P$ but has no $R$-successor; then $\delta_1$ matches the NO labeling exactly, while $\phi^\star$ does not. Early pilot experiments showed that replacing random NO worlds with crafted distractor/trap pools produced more informative FO problems, with real NO-fails and less collapse to trivial formulas while remaining solvable for strong models.
>
> 3. Gold formulas. These are planted reference concepts from a curated structural template pool, not claimed globally canonical minima modulo logical equivalence. They are used as stable complexity anchors, and generation-time filters reject cases where obvious simpler shortcuts already explain the sampled worlds (atomic, proper-subformula, or bounded quantifier-free shortcuts). So we do not claim a proof of global optimality; rather, we use these formulas as stable reference anchors after filtering out obvious simpler shortcuts.
>
> 4. Presentation and definitions. We agree that the paper should have pointed much more clearly to the appendix examples already present there, and that more of the detailed benchmarking can move to the appendix, with the main text focusing on the benchmark setup and the central empirical takeaways. We also agree that more terms should have been defined explicitly in the main text, using more formal definitions where appropriate. On your specific terms: “uniformly across worlds” means that the same single formula must explain all worlds in an instance, not a descriptive-complexity notion of uniformity over all finite structures; “high-capacity models” means stronger frontier models that can satisfy the observed constraints with very large formulas; “guard predicates” are unary guard literals that constrain the choice of an existential witness inside a template family (e.g., $P(y)$ in $\exists y (P(y)\land \forall z (\neg R(y,z)\lor S(x,z)))$); and “lift-hard” refers to subfamilies where the universal body contains an $x$-$z$ lift edge, typically in cross-relational patterns such as $\exists y(P(y)\land \forall z (\neg R(y,z)\lor S(x,z)))$.
>
> 5. Quantifier depth. In the paper this means maximum quantifier nesting depth, i.e. quantifier rank, not alternation count.
>
> We are grateful for this review and hope these clarifications will address the presentation issues you identified and make the paper clearer.

---

> > ### Author Rebuttal · Reviewer_xyir · 2026-04-01
> >
> > I am satisfied with the authors' responses and detailed answers to my questions. If accepted (and I do hope that is the case), please be more precise with definitions, perhaps using the latex definition environment, and include the minimal example.

---

### Official Review · Reviewer_4Nhv · 2026-03-09

**Soundness:** 2
**Presentation:** 4
**Significance:** 4
**Originality:** 3
**Overall Recommendation:** 4
**Confidence:** 4

**Summary:**

The paper studies the ability of AI systems to synthesize first order definitional concepts. Specifically, they introduce INDUCTION,  which provides a methodology for building logic benchmarks. The benchmarks contain fully specified, mechanically verifiable, first order semantical worlds with extensionally specified known predicates and target predicate. In a broader sense, this falls under evaluation of (first order) concept learning or logic program synthesis (not restricted to horn clauses, and without recursion), with a focus on generalization of logical structure. Notable differences include finite verifiability, as well as learning across multiple worlds with the same signature, but potentially different interpretations of known predicates. The authors point out that existing evaluation are influenced by data set artifacts, unverifiable free-form answers etc. The authors present the results of various experiment and broadly conclude that generalization was better when succinct hypotheses were found, which is in line with the standard inductive bias of simplicity.
The worlds contain unary and binary predicates without equality. The worlds were designed to verify how well possible failure modes were handled. Specifically, the benchmark provides 3 scenarios – fully observable worlds (each world has known predicates and a target predicate to be learned – the failure mode tested is overfitting, specifically learning very lengthy formulae with case structures), contrastive learning (worlds split into ‘true’ and ‘false’ worlds where the target formula, and hence the learned formula is true in true worlds and false in false worlds – the false worlds are designed to give information and the failure mode tested is the inability to use negative information) and existential closure (worlds have known predicates that are not fully specified, and the target concept may use the unspecified predicate to have a particular value – failure mode checked is reasoning under missing information.  Further, the target predicates can be generated from a pool of hypothesis which also contain several traps – i.e., similar hypothesis which are small mutations of the intended formula for the target.  The worlds are generated to make sure that these distractors/traps are eliminated.  The benchmark also ensures that atomic concepts, quantifier-free concepts are avoided.
The authors performed experiments on existing AI systems testing these failure modes on holdouts. One thing the authors focus on is the size of the learned formula. Since the target hypothesis are known, the learned formula can be compared in size and other complexity measures such as nesting of quantifiers. The authors show results conforming the idea that formula complexity is a good indicator of generalization.

**Compliance With Llm Reviewing Policy:**

Affirmed.

**Key Questions For Authors:**

1. why was the generation of target formula not exhaustive?
2. can you provide a more comprehensive error analysis? - which formulas were missed and how?

**Limitations:**

yes

**Strengths And Weaknesses:**

Strength:
  This is an important area of study and the authors attempt a systematic testing and give a methodology of bookmark.

Soundness: There are two "claims" - (i) benchmark created (ii) evaluation of existing systems.
Benchmark:
Given the size of models, which relate to the number of target formula, the authors do not justify why a probabilistic generation method was chosen instead of one exhaustive one - i.e., systematic, exhaustive generation of gold standard target. o
gold standard formula - the authors do not specify if these formula are the "simplest" wrt their complexity measure(s), i.e., of the many logically equivalent formulae representing the target concept, is the this the simplest? How many logically equivalent formulae have similar  complexity measures?
Evaluation:
The paper would benefit from results, especially graphs,  that add granularity to the complexity of formulae
Another similar addition is an example gold standard and answers recovered within the paper.
A more important detail that is needed is an error analysis - e.g., the authors notice the free variable of the target occurring inside a universal quantifier makes it harder. How does existential quantizer perform? Is it harder if the target variable occurs (only) in the premise of an implication within a universal quantifier? These analysis are not arbitrary - they correspond to known difficulties in the filled of ILP. Eg: learning an invented predicate is hard. Though this benchmark does not allow for invented predicates, it is direct to inline the invented predicates. Are such formulae harder? A related question is when mistakes are made, do they consistently result in specific types of mistakes? One of the major advantages of a logical domain is that such analyses can be done.

The authors provide an example of prompt, and specify that generalization is important. It would be a good followup to prompt the system to provide small formulae.

---

> ### Author Rebuttal · Authors · 2026-03-30
>
> Thank you for the careful and technically engaged review. Your suggestion to add a more structural error analysis materially strengthens the paper.
>
> We chose a curated template library rather than random formulas because the benchmark needs controlled structural coverage, calibrated difficulty, natural distractors, and interpretable, diagnostically useful error analysis. Random formulas would oversample degenerate or uninformative concepts and make it harder to build worlds that test the intended structure rather than accidental shortcuts. The central challenge is therefore informative-world construction rather than formula enumeration: in FullObs, each accepted world must shrink the survivor set; in CI, YES worlds preserve plausible shortcuts and NO worlds eliminate them; and we filter cases where atomic, subformula, or bounded quantifier-free shortcuts already solve the sampled worlds.
>
> We also agree that the paper should show at least one worked gold formula together with representative model outputs in the main text.
>
> On gold formulas: they are planted reference concepts from a curated template pool, not claimed globally canonical minima modulo logical equivalence. They serve as stable complexity anchors, with filters removing cases where obviously simpler formulas already explain the sampled worlds.
>
> Following your suggestion, we ran a new structural breakdown based on quantifier profiles, the placement of $x$ inside universal rules, and the structure of returned formulas. We also analyzed inlined helper-like structures: formulas that effectively inline an auxiliary predicate by introducing a witness and then imposing a second relational condition through it. The main finding is that hardness comes from universal relational generalization rather than just quantification, especially when a witness must satisfy a universally quantified constraint on further neighbors.
>
> Representative findings:
>
> - Pure existential formulas are much easier than mixed existential/universal formulas. In FullObs, adding universal structure causes a large drop: GPT-5.2 falls by 31 points (41.1% to 9.8%) and Grok4 by 39 points (73.8% to 35.1%); in EC, GPT-5.2 falls by 68 points (93.2% to 25.3%) and Gemini 3 by 76 points (92.0% to 16.5%).
>
> - When $x$ appears only in the premise of a universal rule, performance also drops sharply. On this slice in EC, GPT-5.2 falls by 56 points (76.8% to 21.0%), Gemini 3 by 61 points (71.0% to 9.7%), and Grok4 by 35 points (63.8% to 29.0%). A representative hard formula is $\forall y(R(x,y)\to \exists z (S(y,z)\land P(z)))$, where $x$ is positive only if every $R$-neighbor has an $S$-successor satisfying $P$. This formula appears in benchmark problems in all three tasks, and in each task there are instances of it that all three strong models miss.
>
> - Inlined helper-like structure is also hard. In EC, GPT-5.2 falls by 65 points (74.2% to 8.9%), Gemini 3 by 64 points (66.5% to 2.2%), and Grok4 by 57 points (65.8% to 8.9%). A representative example is $\exists y \exists z (R(x,y)\land S(y,z)\land \neg P(z))$. For this exact formula in FullObs, GPT-5.2 is never budget-compliant across its five occurrences (three valid-but-bloated, two invalid), Gemini 3 is invalid in all five, and Grok4 is invalid in four of five.
>
> The returned formulas reveal a consistent error pattern. On hard universal slices, models often collapse global universal conditions into local existential witnesses. For gold formulas with universal structure in EC, among failed or over-budget predictions, the returned formula is pure existential 81.1% of the time for GPT-5.2, 73.9% for Gemini 3, and 70.4% for Grok4. A typical shift is from $\forall y (R(x,y)\to \exists z (S(y,z)\land P(z)))$ to a local clause such as $\exists y (R(x,y)\land P(y))$.
>
> Models also fail differently on these slices. On the hard nested EC slice, GPT-5.2 still reaches 42.2% raw validity but only 8.9% Acc@+25, indicating many valid-but-bloated surrogates; Gemini 3 is at 4.4% validity and 2.2% Acc@+25, and Grok4 at 8.9% and 8.9%, indicating more outright failure.
>
> We also have additional results that we cannot fit here, including performance by outermost quantifier type, lift-hardness, and recurring error splits such as "YES-fail" versus "NO-fail" in CI. In revision we will add: (i) structural performance tables by gold-formula features and representative features of model-returned formulas; (ii) recurring error types on the hard slices; (iii) a worked gold/prediction example in the paper; and (iv) fuller worked examples for FullObs, CI, and EC in the appendix.
>
> We also agree with your suggestion about smaller formulas. The prompts already encourage compactness and discourage redundancy; EC is more explicit about minimizing AST size, so a compactness-prompt ablation is a natural next step.
>
> We hope these new structural findings and planned appendix additions address the main concerns, and we would appreciate reconsideration of the score in light of them.

---

> > ### Author Rebuttal · Reviewer_4Nhv · 2026-04-01
> >
> > The authors did not address one concern:
> >    I was not asking for randomly generated target formulae but an exhaustive generation of target formulae.
> > I am happy with the other answers.
> > I think the authors should include these results in their paper

---

> > > ### Author Response · Authors · 2026-04-04
> > >
> > > Thank you for clarifying this point. You are right that your remaining concern is specifically about exhaustive generation of target formulas, and our previous response spoke more about informative world construction than about the target-generation procedure itself.
> > >
> > > The clearest way to answer this is to separate two levels of generation. We do not claim exhaustive syntactic enumeration of all first-order formulas under a raw size/depth bound. Instead, the benchmark first fixes a bounded target language, defined by a structured template library chosen to give balanced coverage of the logical patterns the benchmark is meant to test. We made that design choice because exhaustive enumeration over all bounded syntax would be dominated by syntactic near-duplicates (renamings, Boolean rewrites, argument-order variants, etc.) and therefore would not produce a balanced or diagnostically useful benchmark distribution.
> > >
> > > Once that bounded target language is fixed, however, enumeration is exhaustive within it at the eligible-pool level. Concretely, we:
> > > 1. Start from the structural template library;
> > > 2. Expand it into concrete candidate targets by instantiating predicate choices and argument-order variants, with deduplication;
> > > 3. Apply the task- and band-specific complexity and structural filters;
> > > 4. Exhaustively scan the resulting eligible target pool.
> > >
> > > The released benchmark is then not an exhaustive listing of all eligible targets. Rather, it is a balanced sampled subset of that exhaustively scanned pool, followed by stochastic construction of informative world sets and rejection of instances where simpler shortcuts already solve the sampled worlds.  Thus the randomness enters only after the exhaustive scan step.
> > >
> > > So the most accurate characterization of v1 is: not exhaustive over all bounded first-order syntax, but systematic and exhaustive at the eligible-pool level within the benchmark's bounded target language. More precisely, for each task or band, the generator exhaustively scans all formulas in the eligible pool obtained from that bounded target language after applying the relevant complexity and structural restrictions. The released benchmark is then not a complete listing of all such eligible formulas, but a balanced sampled subset of that exhaustively scanned pool. The same distinction applies across FullObs, CI, and EC.
> > >
> > > If a concrete scale is helpful: the released benchmark represents only a balanced subset of the eligible formulas scanned for each task. This is about 40% in FullObs and about 40--50% in the main CI and EC slices. These percentages refer to concrete eligible formulas obtained by expanding the abstract template library and then filtering to the formulas allowed for a given task or difficulty band under its configured complexity and structural restrictions.
> > >
> > > We agree that the paper did not explain this clearly enough. The current text states that targets come from a curated bounded template library and that Appendix~B lists the main structural families and templates, but it does not spell out the full target-generation algorithm or the distinction between the structural template library and the larger task-specific eligible pools obtained after predicate/argument expansion and filtering.
> > >
> > > In the revised appendix, we will make this explicit by adding: (i) a precise description of target-pool construction, (ii) the exhaustive scan and final gold-selection procedure, and (iii) a pool-size census before and after task-specific filtering. We will also include the new structural analyses and worked examples from our previous response, as you suggested.
> > >
> > > Thank you again for pressing us on this point. We believe this clarification will make the generation methodology much easier to understand, and we hope it addresses your remaining concern.

---

### Official Review · Reviewer_vHnJ · 2026-03-11

**Soundness:** 3
**Presentation:** 2
**Significance:** 3
**Originality:** 3
**Overall Recommendation:** 4
**Confidence:** 4

**Summary:**

The paper introduces a benchmark for evaluating LLMs and other systems on a class of inductive logical problemas: learning a first-order formula  with a single free variable, also called a concept or a unary query,  such that in different relational structures the free variable binds to positive instances and does not bind to negative instances. The inductive logical problem is related to the one associated with inductive logic programming, where the formulas must have a particular form, and more general to the concept learning problem, also called, the bounded fitting problem in description logics. DIfferent LLMs are evaluated on this formal inductive task, and they are compared. A big part of the paper is about the construction of the benchmark set.

**Compliance With Llm Reviewing Policy:**

Affirmed.

**Final Justification:**

The authors' rebuttal addressed my concerns, and I hope that the final version of the paper is extended with good, simple, meaningful examples, interleaved with the defintitions, and the symbolic baselines, that illustrate that the benchmark is challenging but not impossibly difficult

**Key Questions For Authors:**

1. How the LLMs do on the proposed task? Can this be described in a clear, compact, and faithful manner? I see the plots but the terminology gets in the way to get an answer to this question.

2. Wouldn't be useful to have some formal solver as a baseline? This is indeed a very hard computationally task.

3. What about humans? Wouldn't it give extra meaning to the result to have results from humans? Humans can also do the task but their performance is also likely to be influenced by the concrete predicates and constant names, and even in the way in which the information is presented (e.g., finding referring expressions for objects in images, etc).

**Limitations:**

Yes

**Strengths And Weaknesses:**

A stength of the paper is that it proposes a crisp learning test to determine whether LLMs do reasoning or mostly do associations.

The weakness, in my view, include that: 1) the task is computationally hard, 2) not formal solvers are used as baselines, which could provide useful information about the hardness of the tasks, 3) it is a formal task, where the name of the predicates and constants should not matter, yet this aspects is completely bypassed in the paper. Very likely, the performance of the LLMs would be affected by the particular predicate and object names used, and also the prompting strategy. This fragility is also a problem for LLMs attempting to do reasoning.

Also, I cannot see clearly the conclusions drawn from the experiments that are expressed in very technical terms. Can the LLMs solve the proposed task, in the given benchmarks, in a reliable way, when the task is "easy enough" as measure by some parameters. Part of this is probably is reflected in the notions like the Delta's. but I'm not sure that I grasped this (and I know logic and I'm probably better position to grasp this than most readers).

---

> ### Author Rebuttal · Authors · 2026-03-30
>
> Thank you for the careful review that helps us strengthen the manuscript. We agree with several important issues you raised, especially the need for stronger symbolic baselines and a clearer plain-language statement of the paper's empirical conclusions.
>
> A clearer summary is: frontier models show real but incomplete capability on these tasks, and not reliable overall performance. They can often fit the observed worlds in easier CI and EC settings, while robust compact multi-world induction remains difficult, especially in FullObs, and formulas much longer than the planted target generalize substantially worse. Accordingly, the paper reports both raw validity and budgeted, bloat-aware success rather than exact-match accuracy alone. For example, in FullObs, lift-hard instances remain much harder than non-lift ones: at Acc@+25, GPT-5.2 drops from 23.3% to 0.0%, Gemini 3 from 18.2% to 0.0%, and Grok4 from 53.2% to 13.1%.
>
> We also agree that symbolic baselines are essential here, and this was an important omission in the original submission. We have now completed semantics-checked symbolic baselines under a common protocol: every symbolic output is re-evaluated by the benchmark checker under exact task semantics, with a 30-minute timeout per instance: SMT-based formula synthesis, grammar-based synthesis, and ILP/ASP-style rule learning, described more fully in our response to Reviewer ogh5. These baselines cover the main plausible symbolic families.
>
> The unified SMT baseline, which synthesizes bounded prenex first-order formulas across all three tasks, achieves FullObs 183/375 = 48.8%, CI 164/200 = 82.0%, and EC 200/200 = 100.0%; a stronger FullObs-specific Z3 portfolio reaches 247/375 = 65.9% on FullObs; and restricted-fragment ILASP reaches 59/375 = 15.7% on FullObs and 91/200 = 45.5% on CI. To our knowledge, no off-the-shelf system is a perfect match to this benchmark, which is not standard Horn-rule induction or generic synthesis: it requires exact finite-world task semantics, including contrastive YES/NO evaluation in CI and existential-completion semantics in EC. As a result, we implemented translated symbolic baselines based on z3 rather than relying on a pre-existing package.
>
> These new rows materially sharpen the calibration story. INDUCTION is paradigm-agnostic as a benchmark, while the paper's scientific motivation is to test and track progress of modern general-purpose models, especially LLMs, on exact first-order reasoning. In the new solver results, the fair unified SMT row is strong on CI and EC but substantially weaker on FullObs, while a more engineered FullObs-specific portfolio closes much of that gap. So the remaining FullObs gap is largely between generic unified search and task-specific symbolic engineering. Further engineering, or more extensive generic search, are likely to cover a significant fraction of the remaining unsolved cases.
>
> On why LLMs are studied here at all: INDUCTION is a paradigm-agnostic benchmark. We do not evaluate LLMs to suggest they should replace solvers. Instead, we aim to study their widely claimed general reasoning capabilities. If LLMs are to act as general reasoning systems, we must measure whether they can recover concise, mechanically checkable symbolic concepts from relational evidence.
>
> We also took your naming-fragility concern seriously and ran a paired 30-instance symbol-renaming pilot (10 FullObs, 10 CI, 10 EC), consistently renaming predicates and objects throughout the task text and instance data: P -> Foo, Q -> Bar, R -> Blorp, S -> Wump, and a_i -> dax_i. Performance was broadly stable rather than collapsing: overall accuracy changed from 40.0% to 33.3% for Gemini 3 and from 66.7% to 73.3% for GPT-5.2, with 100% parse rate throughout. CI and EC were essentially unchanged on this small paired sample, while the 10-instance FullObs slice showed higher variance. The GPT-5.2 increase (+2 problems) and Gemini 3 decrease (-1 problem) are likely due to pilot-scale variance, and the main point is that performance does not collapse under renaming or depend on standard logical notation. Prompt sensitivity is a separate robustness axis, and we agree it merits follow-up study.
>
> We agree that a human baseline would be meaningful. However, a fair human study here is not lightweight: the task is time-intensive even for trained participants, and meaningful measurement would require a carefully designed interface, clear instructions, and enough time budget to separate reasoning difficulty from presentation effects. We therefore saw a human baseline as a valuable follow-up, but out of scope for the initial benchmark paper.
>
> We hope these additions address the main concerns, and we would appreciate reconsideration of the score in light of them.

---

> > ### Author Rebuttal · Reviewer_vHnJ · 2026-04-01
> >
> > The rebuttal clarifies the performance of symbolic solvers on the proposed benchmark. It'd be useful to provide sufficient details on these encodings and solvers. In particular, you say "The unified SMT baseline, which synthesizes bounded prenex first-order formulas across all three tasks .." does best. What's actually the reference for this baseline? In the final version, please also include simple, meaningful examples to illustrate the different logical inductive tasks.

---

### Official Review · Reviewer_ogh5 · 2026-03-12

**Soundness:** 3
**Presentation:** 3
**Significance:** 3
**Originality:** 2
**Overall Recommendation:** 5
**Confidence:** 4

**Summary:**

This paper introduces INDUCTION, a benchmark for evaluating whether models can synthesize correct and concise first-order logic formulas from relational data. Given several finite relational worlds with predicates and an extensional target concept $T(x)$, the task is to generate a formula $\varphi(x)$ that correctly explains the concept across worlds, with correctness verified by exact model checking. Experiments show that although some models can produce valid formulas, many rely on overly complex expressions that overfit training worlds, while simpler formulas generalize significantly better, revealing a gap between validity and true conceptual understanding.

**Compliance With Llm Reviewing Policy:**

Affirmed.

**Final Justification:**

My concerns have been fully addressed. I am satisfied with the comparison on classical symbolic baselines.

**Key Questions For Authors:**

How well do traditional symbolic methods, such as inductive logic programming (ILP), perform on the proposed dataset?

What is the necessity of using large language models for this type of problem, and what advantages do they provide over traditional symbolic methods?

**Limitations:**

Yes.

**Strengths And Weaknesses:**

1. The paper presents a clear motivation for studying whether modern language models can genuinely perform symbolic concept induction rather than relying on superficial patterns. The task formulation is precisely defined with solver-verifiable semantics, which makes correctness objectively measurable. The dataset generation pipeline appears carefully designed, including mechanisms such as hypothesis pools, trap construction, and filtering rules to avoid trivial solutions. Overall, the benchmark provides a meaningful and controlled setting for evaluating the symbolic reasoning capabilities of LLMs.

2. The evaluation protocol is thoughtfully designed and goes beyond simple accuracy. In particular, the paper introduces metrics that explicitly account for the syntactic complexity of the generated formulas, such as AST size and budgeted accuracy relative to the ground-truth formula. This design is especially innovative in the context of first-order logic synthesis, where multiple logically correct but overly complex formulas can exist.

3. While the benchmark focuses on evaluating modern language models, the paper does not include comparisons with traditional symbolic learning approaches, such as inductive logic programming (ILP) systems or program synthesis methods that operate over logical representations. Including such baselines could provide additional insight into the difficulty of the tasks and help contextualize the performance of LLM-based approaches. Without these comparisons, it is harder to assess whether the benchmark primarily challenges neural models or whether it is broadly difficult for existing symbolic learning techniques as well.

---

> ### Author Rebuttal · Authors · 2026-03-30
>
> Thank you for your careful review. We appreciate that you found the benchmark design meaningful and highlighted the value of solver-verifiable semantics and compactness-aware evaluation.
>
> We agree that symbolic baselines are essential for contextualizing the benchmark, and this was an omission in the original submission. We have now completed symbolic baselines under a common protocol: every symbolic hypothesis is evaluated by the benchmark's own finite-world checker under the exact task semantics, with a 30-minute timeout per instance. We chose that budget because it is comparable to frontier model runtimes on harder instances.
>
> Symbolic rows (FullObs / CI / EC):
> - z3-prenex + generic qd2/d3 rescue: 183/375 (48.8%) / 164/200 (82.0%) / 200/200 (100.0%)
> - z3-mix: 247/375 (65.9%) / -- / --
> - ilasp-frag: 59/375 (15.7%) / 91/200 (45.5%) / --
>
> Method notes:
> - z3-prenex is the fair unified SMT baseline across all three tasks. It synthesizes prenex formulas over the benchmark predicates; for EC, it uses shared per-world completion variables so the encoding matches existential-completion semantics exactly. On FullObs, we add only a generic second pass on unsolved cases, restricted to two empirically productive qd2-depth-3 families.
> - z3-mix is a stronger FullObs-only Z3 portfolio. Its advantage comes from encoding the dominant FullObs qd2 patterns as much smaller SMT problems than unrestricted formula search, so we report it as a task-specific row rather than as a fair generic baseline.
> - ilasp-frag is a restricted ILASP encoding whose outputs are decoded back to first-order formulas and re-checked by the benchmark evaluator, so the reported numbers reflect valid benchmark hypotheses rather than raw ILASP outputs.
> - We also explored a SyGuS/cvc5 baseline, but the stable formulation was much weaker than the Z3 baselines and a cleaner revised formulation was not robust enough for benchmark-scale reporting.
>
> Together, these baselines cover the main symbolic families plausibly applicable here: SMT-based formula synthesis, grammar-based synthesis, and ILP/ASP-style rule learning. At the same time, no off-the-shelf system is a perfect match to this benchmark, which is not standard Horn-rule induction or generic synthesis: it requires exact finite-world task semantics, including contrastive YES/NO evaluation in CI and existential-completion semantics in EC. That is why we implemented several translated symbolic baselines rather than relying on a single package.
>
> These additions substantially strengthen the calibration story. The benchmark is not LLM-only, and it is meaningful for classical symbolic methods as well. At the same time, there is no single trivial symbolic row that solves all three regimes cleanly: the fair unified SMT row is strong on CI and EC but substantially weaker on FullObs, while a more engineered FullObs-specific portfolio closes much of that gap. This should not be read as a hard symbolic ceiling. It is a bounded search under a fixed compute budget, and broader symbolic search would predictably trade more compute for more coverage. The benchmark also yields concrete structural findings. For example, in FullObs, lift-hard instances are dramatically harder than non-lift ones: at Acc@+25, GPT-5.2 drops from 23.3% to 0.0%, Gemini 3 from 18.2% to 0.0%, and Grok4 from 53.2% to 13.1%; we summarize a broader structural breakdown in our response to Reviewer 4Nhv. More generally, the benchmark separates valid-but-bloated solutions from compact ones that generalize, which is exactly what its evaluation protocol is designed to surface.
>
> On your question about why LLMs are studied here at all: INDUCTION is a paradigm-agnostic benchmark. We do not evaluate LLMs to suggest they should replace solvers, but to rigorously test their widely claimed general reasoning capabilities. If LLMs are to act as general reasoning systems, we must measure whether they can recover concise, mechanically checkable symbolic concepts from relational evidence. The relevant advantage, if any, is not task-specific efficiency here, but potential generality across tasks without hand-engineered symbolic search bias. Symbolic baselines are complementary: they show whether the benchmark is broadly difficult and how current LLM behavior compares with classical search-based learners.
>
> More broadly, we would frame the paper's contribution as the combination of: (i) a unified, solver-verifiable benchmark for cross-world FO concept induction, (ii) three complementary regimes targeting different failure modes (FullObs, CI, EC), and (iii) budgeted, gold-relative evaluation that separates valid-but-bloated solutions from compact solutions that generalize. The new symbolic results strengthen that contribution rather than changing it.
>
> We hope these new results address the main concern about evaluation breadth, and we would appreciate reconsideration of evaluation of the paper in light of the strengthened cross-paradigm comparison.

---

> > ### Author Rebuttal · Reviewer_ogh5 · 2026-04-03
> >
> > My concerns have been fully addressed.

---

### Decision · Program_Chairs · 2026-04-30

**Decision:**

Accept (spotlight)

**Comment:**

The paper presents a benchmark for three reasoning tasks in first-order logic, all verifiable with tools such as SMT solvers.  While some of the tables report high accuracy, the contribution is interesting because it’s always possible to increase the complexity of the formulas. The discussion provided the authors with good advice on how to improve the paper, including adding symbolic baselines and examples.

I strongly recommend this paper to be accepted. It's a well-rounded paper. It needs substantial work, but I see it is likely that the authors can accomplish it.